# ON THE FAIRNESS ROAD: ROBUST OPTIMIZATION FOR ADVERSARIAL DEBIASING

**Vincent Grari**[*,1,2,4], **Thibault Laugel**[*,1,2,4], **Tatsunori Hashimoto**[2], **Sylvain Lamprier**[3], **Marcin Detyniecki**[1,4,5]

[1] AXA Group Operations
[2] Stanford University
[3] LERIA, Université d'Angers, France
[4] TRAIL, Sorbonne Université, Paris, France
[5] Polish Academy of Science, IBS PAN, Warsaw, Poland

`{grari,laugel}@stanford.edu`
`code: https://github.com/axa-rev-research/ROAD-fairness/`

## ABSTRACT

In the field of algorithmic fairness, significant attention has been put on group fairness criteria, such as Demographic Parity and Equalized Odds. Nevertheless, these objectives, measured as global averages, have raised concerns about persistent local disparities between sensitive groups. In this work, we address the problem of local fairness, which ensures that the predictor is unbiased not only in terms of expectations over the whole population, but also within any subregion of the feature space, unknown at training time. To enforce this objective, we introduce ROAD, a novel approach that leverages the Distributionally Robust Optimization (DRO) framework within a fair adversarial learning objective, where an adversary tries to predict the sensitive attribute from the predictions. Using an instance-level re-weighting strategy, ROAD is designed to prioritize inputs that are likely to be locally unfair, i.e. where the adversary faces the least difficulty in reconstructing the sensitive attribute. Numerical experiments demonstrate the effectiveness of our method: it achieves, for a given global fairness level, Pareto dominance with respect to local fairness and accuracy across three standard datasets, as well as enhances fairness generalization under distribution shift.

## 1 INTRODUCTION

The increasing adoption of machine learning models in various applications such as healthcare or criminal justice, has raised concerns about the fairness of algorithmic decision-making processes. As these models are often trained on historical data, they have been shown to unintentionally perpetuate existing biases and discrimination against certain vulnerable group (Obermeyer et al., 2019). Addressing fairness in ML has thus become an essential aspect of developing ethical and equitable systems, with the overarching goal of ensuring that prediction models are not influenced by sensitive attributes. One of its most common concepts, group fairness, entails dividing the population into demographic-sensitive groups (e.g., male and female) and ensuring that the outcomes of a decision model are equitable across these different groups, as measured with criteria like Demographic Parity (DP) (Dwork et al., 2012) and Equal Opportunity (EO) (Hardt et al., 2016).

However, focusing solely on these group fairness criteria, along with predictive performance, has been increasingly questioned as an objective: besides being shown to poorly generalize to unseen, e.g. drifted, environments (Kamp et al., 2021), it has been more generally criticized for being too simplistic (Selbst et al., 2019; Binns, 2020), leading to arbitrariness in the bias mitigation process (Krco et al., 2023) and the risk of having some people pay for others (Mittelstadt et al., 2023). Recognizing these issues, some researchers have long focused on exploring more localized fairness behaviors, proposing to measure bias sectionally within predefined demographic categories, in which comparison between sensitive groups is deemed meaningful for the considered task. For instance, using *Conditional Demographic Disparity* (Žliobaite et al., 2011), fairness in predicted

---

*Equal contribution

salaries between men and women shall be evaluated by comparing individuals within the same job category and seniority level, rather than making a global comparison across sensitive groups.

Nevertheless, predefining these *comparable* groups to optimize their local fairness is often difficult: for instance, which jobs should be deemed legally comparable with one another? (Wachter et al., 2021) In this paper, we therefore propose to address the difficult problem of enforcing fairness in local subgroups that are unknown at training time (Sec. 2). For this purpose, we leverage the Distributionally Robust Optimization (DRO) framework, initially proposed to address worst-case subgroup accuracy (see e.g. Duchi & Namkoong (2021)). Our approach ROAD (*Robust Optimization for Adversarial Debiasing*, described in Sec.3) combines DRO with a fair adversarial learning framework, which aims to minimize the ability of an adversarial model to reconstruct the sensitive attribute. By boosting attention on feature regions where predictions are the most unfair in the sense of this sensitive reconstruction, ROAD is able to find the best compromise between local fairness, accuracy and global fairness. Such dynamic focus is done by relying on a weighting process that respects some locality smoothness in the input space, in order to mitigate bias in any implicit subgroup of the population without supervision. Experiments, described in Section 4, show the efficacy of the approach on various datasets.

## 2 PROBLEM STATEMENT

Throughout this document, we address a conventional supervised classification problem, trained using $n$ examples $(x_i, y_i, s_i)_{i=1}^n$, where each example is composed of a feature vector $x_i \in \mathbb{R}^d$, containing $d$ predictors, a binary sensitive attribute $s_i$, and a binary label $y_i$. These examples are sampled from a training distribution $\Gamma = (X, Y, S) \sim p$. Our goal is to construct a predictive model $f$ with parameters $w_f$ that minimizes the loss function $\mathcal{L}_\mathcal{Y}(f(x), y)$ (e.g. log loss for binary classification), whilst adhering to fairness constraints based on specific fairness definitions relying on the sensitive attribute $S$. In this section, we present the fairness notions and works that are necessary to ground our proposition.

### 2.1 GROUP FAIRNESS

One key aspect of algorithmic fairness is group fairness, which aims to ensure that the outcomes of a decision model are equitable across different demographic groups. In this paper, we focus on two of the most well-known group fairness criteria: *Demographic Parity* and *Equalized Odds*.

*Demographic Parity*: Demographic parity (DP) (Dwork et al., 2012) is achieved when the proportion of positive outcomes is equal across all demographic groups. Using the notations above, the learning problem of a model $f$ under demographic parity constraints can be expressed as follows:

$$\underset{w_f}{\arg\min} \quad \mathbb{E}_{(x,y,s) \sim p} \, \mathcal{L}_\mathcal{Y}(f_{w_f}(x), y) \quad \text{s.t.} \quad |\mathbb{E}_p[\hat{f}_{w_f}(x)|s=1] - \mathbb{E}_p[\hat{f}_{w_f}(x)|s=0]| < \epsilon \quad (1)$$

Where $\hat{f}$ represents the output prediction after threshold (e.g., $\hat{f}_{w_f}(x) = \mathbb{1}_{f_{w_f}(x)>0.5}$). The parameter $\epsilon$ represents the deviation permitted from perfect statistical parity, allowing for flexibility in balancing accuracy and fairness. In the following, this deviation is noted as *Disparate Impact* (DI), representing the absolute difference in positive outcomes between the two demographic groups.

Although numerous methods exist to solve the problem described in Equation 1, we focus in this work on the family of fair adversarial learning, which has been shown to be the most powerful framework for settings where acting on the training process is an option (i.e., in-processing method) (Louppe et al., 2017; Wadsworth et al., 2018; Zhang et al., 2018; Grari, 2022). One of the most well-known fair adversarial approaches by Zhang et al. (2018) is framed as follows:

$$\min_{w_f} \quad \mathbb{E}_{(x,y,s) \sim p} \, \mathcal{L}_\mathcal{Y}(f_{w_f}(x), y) \quad \text{s.t.} \quad \min_{w_g} \mathbb{E}_{(x,y,s) \sim p} \mathcal{L}_\mathcal{S}(g_{w_g}(f_{w_f}(x)), s) > \epsilon' \quad (2)$$

Where $\mathcal{L}_\mathcal{S}$ represents a loss for sensitive reconstruction (e.g. a log loss for a binary sensitive attribute). In this adversarial formulation, the goal is to learn a model $f$ that minimizes the traditional loss of the predictor model, while simultaneously ensuring that an adversary $g$ with parameters $w_g$ cannot effectively distinguish between the two sensitive demographic groups based on the predictor's output $f_{w_f}(x)$. The fairness constraint is thus imposed here as the adversary's

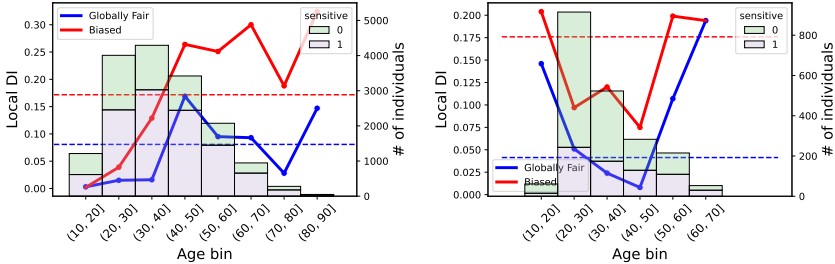

Figure 1: Visualizing local fairness: DI values measured globally (dashed lines) and locally (full lines) for a biased model (red) and one trained with a DI objective (in blue) (Zhang et al. (2018) with $\lambda$=3). The bar plots show the sensitive attribute distribution in each age group. Left: Adult dataset; Right: Compas.

ability to reconstruct the sensitive attribute, which should be limited, i.e. the value of the loss function $\mathcal{L}_{\mathcal{S}}(g_{w_g}(f_{w_f}(x)), s)$ should be above a minimum value $\epsilon'$. In practice, to achieve a balance between the predictor's and the adversary's performance, a relaxed formulation of Equation 2 is used: $\min_{w_f} \max_{w_g} \quad \mathbb{E}_{(x,y,s) \sim p} \mathcal{L}_{\mathcal{Y}}(f_{w_f}(x), y) - \lambda \mathbb{E}_{(x,y,s) \sim p} \mathcal{L}_{\mathcal{S}}(g_{w_g}(f_{w_f}(x)), s)$. The coefficient $\lambda \in \mathbb{R}^+$ controls the trade-off between the predictor's performance on the task of predicting $Y$ and the adversary's performance on reconstructing the sensitive attribute. A larger value of $\lambda$ emphasizes the importance of restricting the adversary's ability to reconstruct the sensitive attribute, while a smaller value prioritizes the performance of the predictor on the main task.

*Equalized Odds*: Equalized Odds (EO) (Hardt et al., 2016) is another group fairness criterion that requires the classifier to have equal true positive rates (TPR) and false positive rates (FPR) across demographic groups. This criterion is especially relevant when misclassification can have significant impacts on individuals from different groups. To achieve EO, Zhang et al. (2018) employs an adversarial learning approach by concatenating the true outcome $Y$ to the input of the adversary.

## 2.2 THE LOCAL FAIRNESS PROBLEM

The global aspect of these group fairness criteria begs the question of the emergence of local undesired behaviors: by enforcing constraints on global averages between sensitive groups, we still expect that some local differences may persist (Krco et al., 2023). We illustrate this phenomenon through a simple experiment, shown in Fig. 1. On two datasets, Adult and Compas (described in App. A.8.1), two models are trained: an unconstrained model solely optimizing for accuracy (called *Biased*, in red), and the adversarial model from Zhang et al. (2018) (in blue) optimizing for Demographic Parity for the sensitive attributes *gender* (Adult) and *race* (Compas). For each model, two types of Disparate Impact (DI) values are shown: the global DI values, calculated over all the test set (dashed lines); and the local ones, calculated in subgroups of the population (full lines). The subgroups are defined here as *age categories*: discretized bins of the continuous attribute *age*. Although local DI values are generally lower for the fair model, they vary a lot across subgroups, sometimes remaining unexpectedly high. This is especially true for less populated segments (e.g., higher *age* values), and segments where the sensitive attribute distribution is extremely unbalanced: as the fairness constraint only concerns global averages, more attention is put on densely populated regions. On the other hand, less populated segments are more likely to be ignored during the training.

These local differences echo the long-asserted claim that the blunt application of group fairness metrics bears inherent inequalities through their failure to account for any additional context (Selbst et al., 2019; Binns, 2020). Here, although reductive, the additional context we refer to is the information already available in the dataset $X$, in which *comparable* subgroups (Wachter et al., 2021) can be drawn to evaluate fairness. This helps defining the notion of *Local Fairness* that is the focus of this paper: a locally fair model thus guarantees minimal differences in expectations within these comparable subgroups of $X$. Contrary to works on intersectional fairness (Kearns et al., 2018), the desired behavior in Fig. 1 is thus not to treat *age* as a sensitive attribute: predictions $\hat{f}(x)$ are expected to vary along *age*. However, in the Compas dataset for instance, equality between *race* groups is expected to hold regardless of the *age* category considered. It is important to note that the

notion studied here is also different from the one of *individual fairness*, which aims to treat similarly individuals who are close w.r.t. some predefined similarity measure (see, e.g. Dwork et al. (2012)), without any notion of sensitive data, rather than minimize DI among subgroups of individuals. In the same vein of fairness without demographics, (Hashimoto et al., 2018; Duchi et al., 2023) consider the case of unknown subgroups via the Distributionally Robust Optimization (DRO) framework. While their goal is to train models that perform uniformly well across all partitions of the population, our goal is to train a model that is uniformly fair (regarding a sentitive attribute) accross all subregions of the feature space, which is quite different.

Having knowledge of these subgroups at training time would mean that it could be included as an additional constraint in the learning objective, akin to the work of Žliobaite et al. (2011). The criterion they propose, Conditional Demographic Disparity, measures Demographic Disparity across user-defined subcategories. However, several issues make this difficult, if not impossible, in practice. Besides that such expert knowledge is generally unavailable, or costly to acquire, the subgroups definitions might even be inconsistent across different testing environments (e.g. conflicting legal definitions of job categories or gender (Wachter et al., 2021)), making its optimization futile. Furthermore, exploring multiple categories is problematic in a combinatorial perspective. In this paper, we propose to optimize accuracy while adhering to a worst-case fairness constraint, an objective that was originally introduced to enhance fairness generalization capabilities in scenarios involving distribution drift or noisy labels (cf. Sec. 2.3). We implicitly define the subpopulations of interest, for which we aim to optimize fairness, using distributions $q$ within an *uncertainty set* $\mathcal{Q}$, and present the DRO framework for the Demographic Parity criterion as follows:

$$\min_{w_f} \quad \mathbb{E}_{(x,y,s) \sim p} \, \mathcal{L}_{\mathcal{Y}}(f_{w_f}(x), y) \quad \text{s.t.} \quad \max_{q \in \mathcal{Q}} \left| \mathbb{E}_q \left[ \hat{f}_{w_f}(x) | s = 1 \right] - \mathbb{E}_q \left[ \hat{f}_{w_f}(x) | s = 0 \right] \right| < \epsilon \quad (3)$$

The constraint ensures that the Disparate Impact remains less than a predefined threshold $\epsilon$ under the worst-case distribution $q \in \mathcal{Q}$. Working with distribution $q$ allows us to enforce local fairness by targeting subpopulations of interest, thus creating a more focused and adaptable model that addresses fairness problems both globally and at a granular level.

## 2.3 RELATED WORK AND POSITIONING

Several works have proposed to address the objective in Eq 3, either to ensure better fairness generalization capabilities in drift scenarios (Rezaei et al., 2021; Ferry et al., 2022; Wang et al., 2023) or when facing noisy labels (Mandal et al., 2020; Wang et al., 2020; Roh et al., 2021). The uncertainty set $\mathcal{Q}$ then represents the perturbations that might affect the data at test time, and can therefore take several forms. While we expect $\mathcal{Q}$ contains the distribution of test data, leaving too much freedom to $q$ may lead to trivial solutions that degenerate as uniform classifiers (Martinez et al., 2021). To do so, the uncertainty set $\mathcal{Q}$ is commonly defined as a ball centered on $p$ using distribution distances or similarities. Examples include maximal Total Variation distance (Wang et al., 2020), Wasserstein distance (Wang et al., 2021) or Jaccard index (Ferry et al., 2022). From the fairness without demographics literature (Duchi et al., 2023), it is known that the maximal allowed divergence is connected to the risk of the smallest component of the training distribution, seen as a mixture of distributions. This observation also holds for worst-case fairness using DRO, as defined in Eq 3.

To the best of our knowledge, our work is the first one to address the topic of local fairness with unknown subgroups. This different objective implies additional constraints on the set $\mathcal{Q}$ considered in Eq 3. Notably, under our local fairness objective, we also want that the discrepancies of $q$ w.r.t. $p$ are smooth in the feature space, so that the fairness constraint does not increase mitigation on specific disconnected individuals, but rather on local areas of the space. This will guide the design of our approach in the next section.

Moreover, due to the discrete nature of the problem expressed in Eq 3 (the constraint is applied on $\hat{f}$ which is binary), most existing works restrict to linear models (Wang et al., 2020; Rezaei et al., 2020; Mandal et al., 2020; Taskesen et al., 2020), or rule-based systems (Ferry et al., 2022). This allows them to look for analytical solutions using linear programming. Although Rezaei et al. (2021) is an exception in this regard, they suffer from several drawbacks, namely requiring knowledge about the target distribution at train time and about the sensitive attribute at test time. Solving Equation 3 using a wider class of models remains therefore, to the best of our knowledge, unexplored.

# 3    ROAD: ROBUST OPTIMIZATION FOR ADVERSARIAL DEBIASING

## 3.1    FORMALIZATION

To overcome the limitations of previous works, we introduce our proposition to address the fairness generalization problem by combining adversarial optimization and the DRO framework. In order to learn a predictor $f_{w_f}$ that is fair both globally and for any subregion of the feature space, the idea is therefore to boost, at each optimization step, the importance of regions $q$ for which the sensitive reconstruction is the easiest for an optimal adversary $g_{w_g*}$ given the current prediction outcomes. Rewriting the fairness constraint of Equation 3 with an adversary $g_{w_g} : \mathcal{Y} \to \mathcal{S}$, we thus focus on the following problem for Demographic Parity[1]:

$$\min_{w_f} \quad \mathbb{E}_{(x,y,s) \sim p} \, \mathcal{L}_{\mathcal{Y}}(f_{w_f}(x), y)$$
$$\text{s.t.} \quad \min_{q \in \mathcal{Q}} \mathbb{E}_{(x,y,s) \sim q} \mathcal{L}_{\mathcal{S}}(g_{w_g*}(f_{w_f}(x)), s) > \epsilon' \tag{4}$$
$$\text{with } w_g{}^* = \arg\min_{w_g} \mathbb{E}_{(x,y,s) \sim p} \mathcal{L}_{\mathcal{S}}(g_{w_g}(f_{w_f}(x)), s)$$

A major challenge with this formulation is that exploring all possible distributions in $\mathcal{Q}$ is infeasible in the general sense. Worse, modeling distribution $q$ directly over the whole feature space as support is very difficult, and usually highly inefficient, even for $\mathcal{Q}$ restricted to distributions close to $p$. This motivates an adversarial alternative, which relies on importance weighting of training samples from $p$. We therefore restrict $\mathcal{Q}$ to the set of distributions that are absolutely continuous with respect to $p$[2], inspired by Michel et al. (2022) . This allows us to write $q = rp$ , with $r : \mathcal{X} \times \mathcal{S} \to \mathbb{R}^+$ a function that acts as a weighting factor. Given a training set $\Gamma$ sampled from $p$, we can thus reformulate the overall objective, by substituting $q$ with $rp$ and applying its Lagrangian relaxation, as an optimization problem on $r \in \mathcal{R} = \{r \mid rp \in \mathcal{Q}\}$:

$$\min_{w_f} \max_{r \in \mathcal{R}} \frac{1}{n} \sum_{i=1}^n \mathcal{L}_{\mathcal{Y}}(f_{w_f}(x_i), y_i) - \lambda_g \frac{1}{n} \sum_{i=1}^n r(x_i, s_i) \mathcal{L}_{\mathcal{S}}(g_{w_g^*}(f_{w_f}(x_i)), s_i) \tag{5}$$

$$\text{with } w_g^* = \arg\min_{w_g} \frac{1}{n} \sum_{i=1}^n \mathcal{L}_{\mathcal{S}}(g_{w_g}(f_{w_f}(x_i)), s_i)$$

With $\lambda_g$ a regularization parameter controlling the trade-off between accuracy and fairness in the predictor model. In the following, we describe two constraints, inspired from the DRO literature, that we consider to ensure $q$ keeps the properties of a distribution and avoids pessimistic solutions.

**Validity Constraint**    To ensure $q$ keeps the properties of a distribution (i.e., $r \in \mathcal{R}$), previous works in DRO (e.g. Michel et al. (2022)) enforce the constraint $\mathbb{E}_{(x,s) \sim p} r(x, s) = 1$ during the optimization. In the context of local fairness using our adversarial formulation from Eq.5, we argue that this constraint is not sufficient to ensure a safe behavior with regard to the fairness criterion, as it allows disturbances in the prior probabilities of the sensitive (i.e., $q(s) \neq p(s)$). As discussed more deeply in Appendix A.2.2, this may lead to a shift of the optimum of the problem, by inducing a stronger mitigation emphasis on samples from the most populated demographic-sensitive group. To avoid this issue, we propose to further constrain $r$ by considering a restricted set $\tilde{\mathcal{R}} = \{r \in \mathcal{R} \mid rp \in \tilde{\mathcal{Q}}\}$, with $\tilde{\mathcal{Q}} \subset \mathcal{Q}$ such that: $\forall s, q(s) = p(s)$. To achieve this, we rely on the following constraint: $\forall s, \mathbb{E}_{p(x|s)} r(x, s) = 1$. Besides guaranteeing the desired property $q(s) = p(s)$ (proof in Sec. A.2.1), we also note that ensuring these constraints still imply the former one: $\mathbb{E}_{p(x,s)} r(x, s) = 1$, which guarantees that $q(x, s)$ integrates to 1 on its support. We further discuss the benefits of this conditional constraint in Section A.2.3.

**Shape Constraint**    As discussed in Section 2.3, the definition of $\mathcal{Q}$ heavily impacts the desired behavior of the solution. In particular, controlling the shape of the allowed distributions $q$ is especially

---

[1]Adapting our work to EO is straightforward: as described in Sec. 2.1, adapting the adversarial method of Zhang et al. (2018) to the EO task simply requires to concatenate the the true outcome $Y$ to the prediction $f(x)$ as input of the adversarial classifier. The same process can be followed for ROAD.

[2]In the situation where all distributions in $Q$ are absolutely continuous with respect to $p$ all measurable subset $A \subset X \times Y$, all $q \in \mathcal{Q}, q(A) > 0$ only if $p(A) > 0$)

crucial in a setting such as ours, where the focus of the mitigation process is done dynamically. Without any constraint (as proposed by Mandal et al. (2020)), the mitigation could indeed end up focusing on specific points of the dataset where the sensitive reconstruction from $f_{w_f}(X)$ is the easiest, using very sharp distributions $q$ close to a Dirac. This may turn particularly unstable and, more critically, could concentrate the majority of fairness efforts on a relatively small subset of samples. To control the shape of the bias mitigation distribution $q$, we therefore choose to consider $\mathcal{Q}$ as a KL-divergence ball centered on the training distribution $p$. However, similarly to Michel et al. (2022), we do not explicitly enforce the KL constraint (due to the difficulty of projecting onto the KL ball) and instead use a relaxed form. Using previous notations, the KL constraint takes the simple form $\text{KL}(q||p) = \text{KL}(pr||p) = \mathbb{E}_p r \log \frac{pr}{p} = \mathbb{E}_p r \log r$.

The spread of $\mathcal{Q}$ can then be controlled with a temperature weight $\tau$ in the overall optimization process, which can be seen as the weight of a Shannon entropy regularizer defined on discrepancies of $q$ regarding $p$. Setting $\tau = 0$ means that no constraint on the distribution of $r$ is enforced, thus encouraging $r$ to put extreme attention to lower values of $\mathcal{L}_S$. On the other hand, higher values of $\tau$ favors distributions $q$ that evenly spreads over the whole dataset, hence converging towards a classical globally fair model for highest values (cf. Section 2.1). Note that setting this hyperparameter is strongly related to implicitly tuning the size of the smallest subgroup of the population for which we ensure fairness (cf. section 2.3).

**ROAD Formulation**  The overall optimization problem of our Robust Optimization for Adversarial Debiasing (ROAD) framework can thus finally be formulated as (full derivation given in A.1):

$$
\min_{w_f} \max_{r \in \tilde{\mathcal{R}}} \frac{1}{n} \sum_{i=1}^{n} \mathcal{L}_Y(f_{w_f}(x_i), y_i) - \lambda_g [\frac{1}{n} \sum_{i=1}^{n} r(x_i, s_i) \mathcal{L}_S(g_{w_g^*}(f_{w_f}(x_i)), s_i)
$$
$$
+ \tau \underbrace{\frac{1}{n} \sum_{i=1}^{n} r(x_i, s_i) \log(r(x_i, s_i))}_{\text{KL constraint}}] \qquad (6)
$$
$$
\text{with } w_g^* = \arg\min_{w_g} \frac{1}{n} \sum_{i=1}^{n} \mathcal{L}_S(g_{w_g}(f_{w_f}(x_i)), s_i)
$$

## 3.2 Two Implementations for ROAD

### 3.2.1 BROAD: A Non-Parametric Approach

Let us first introduce a non-parametric approach, called Boltzmann Robust Optimization Adversarial Debiasing (BROAD), where each $r(x_i, s_i)$ value results from the inner maximization problem from Eq.13. As described below, this inner optimization accepts an analytical solution, whenever $r$ values respect the aforementioned conditional validity constraints (proof in Appendix A.3).

**Lemma 3.1.** *(Optimal Non-parametric Ratio) Given a classifier $f_{w_f}$ and an adversary $g_{w_g}$, the optimal weight $r(x_i, s_i)$ for any sample from the training set, is given by:*

$$
r(x_i, s_i) = \frac{e^{-\mathcal{L}_S(g_{w_g}(f_{w_f}(x_i)), s_i)/\tau}}{\frac{1}{n_{s_i}} \sum_{(x_j, s_j) \in \Gamma, s_j = s_i} e^{-\mathcal{L}_S(g_{w_g}(f_{w_f}(x_j)), s_j)/\tau}}
$$

With $n_{s_i} = \sum_{i=1}^{n} \mathbb{1}_{s=s_i}$. This expression allows us to set optimal weights for any sample from the training dataset, at no additional computational cost compared to a classical adversarial fairness approach such as Zhang et al. (2018). However, this may induce an unstable optimization process, since weights may vary abruptly for even very slight variations of the classifier outputs. Moreover, it implies individuals weights, only interlinked via the outputs from the classifier, hence at the risk of conflicting with our notion of local fairness. We therefore propose another - parametric - implementation, described in the next section, that improves the process by introducing local smoothness in the fairness weights.

### 3.2.2 PARAMETRIC APPROACH

To introduce more local smoothness in the fairness weights assigned to training samples, we propose an implementation of the $r$ function via a neural network architecture. Our goal is to ensure that groups of similar individuals, who might be neglected in the context of group fairness mitigation (e.g., due to their under-representation in the training population, cf. Fig. 1), receive a similar level of attention during the training process. However, solely relying on adversarial accuracy, as done in BROAD, may induce many irregularities in such groups. The lipschitzness of neural networks can add additional implicit locality smoothness assumptions in the input space, thus helping define the distributions $q$ as subregions of the feature space. Note that, in this approach, the network architecture therefore plays a crucial role in how local the behavior of $r_{w_r}$ will be: more complex networks will indeed tend to favor more local solutions, for a same value of $\tau$. In particular, a network of infinite capacity that completes training will have, in theory, the same behavior as BROAD.

To enforce the conditional validity constraint presented earlier, we employ an exponential parametrization with two batch-level normalizations, one for each demographic group. For each sample $(x_i, y_i, s_i)$ in the mini-batch, we define the normalized ratio as:

$$\forall i, r_{w_r}(x_i, s_i) = \frac{\mathrm{e}^{h_{w_r}(x_i, s_i)}}{\frac{1}{n_{s_i}} \sum_{(x_j, s_j) \in \Gamma, s_j = s_i} \mathrm{e}^{h_{w_r}(x_j, s_j)}}$$

with $h : \mathcal{X} \times \{0; 1\} \rightarrow \mathcal{R}$ a neural network with weights $w_r$. To train ROAD, we use an iterative optimization process, alternating between updating the predictor model's parameters $w_f$ and updating the adversarial models' parameters $w_g$ and $w_r$ by multiple steps of gradient descent. This leads to a far more stable learning process and prevents the predictor classifier from dominating the adversaries. More details are provided in the appendix (see Alg. 1).

## 4 EXPERIMENTS

### 4.1 ASSESSING LOCAL FAIRNESS

In this first experiment, we assess how effective ROAD is for generating predictions that are locally fair for unknown subpopulations, while guaranteeing a certain level of global accuracy and global fairness. For this purpose, we use 3 datasets often used in fair classification, described in Appendix A.8.1: Compas (Angwin et al., 2016), Law (Wightman, 1998) and German Credit (Hofmann, 1994). Each dataset is split into training and test subsets, and the models described below are trained to optimize accuracy while mitigating fairness with respect to a sensitive attribute $S$.

To assess fairness at a local level, various subpopulations chosen among features of $X$, i.e. excluding $S$, are selected in the test set. As an example on the Compas dataset, in which $S$ is *Race*: to create the subgroups, *Age* is discretized into buckets with a 10-year range. These intervals are then combined with the *Gender* feature, identifying 12 distinct subgroups. As measuring DI in segments of low population is highly volatile, we filter out subgroups with less than 50 individuals (see App. A.8.3). These subgroups are unknown at training time, and chosen arbitrarily to reflect possible important demographic subgroups (see Sec. 4.3.2 for further discussion). Given these subgroups $\mathcal{G}$, the local fairness is then assessed on the worst Disparate Impact value across these subgroups: *Worst-1-DI* $= \max_{g \in \mathcal{G}} |\mathbb{E}_{(x,s) \in g}(\hat{f}_{w_f}(x)|s = 1) - \mathbb{E}_{(x,s) \in g}(\hat{f}_{w_f}(x)|s = 0)|$. To evaluate our approach, we compare our results with the globally fair adversarial models from Zhang et al. (2018) and Adel et al. (2019), and 3 approaches that address fairness generalization: FairLR (Rezaei et al., 2020), RobustFairCORELS (Ferry et al., 2022) and CUMA (Wang et al., 2023) (cf. App. A.8.2).

As local fairness can only be measured against global accuracy and fairness, we evaluate the approaches by plotting the tradeoffs between global accuracy and worst-1-DI subject to a global DI constraint (we choose $DI \leq 0.05$, following the fairness literature (Pannekoek & Spigler, 2021)). To ensure a thorough exploration of these tradeoffs, we sweep across hyperparameter values for each algorithm (hyperparameter grids in App. A.8.4). Fig. 2 shows the resulting Accuracy-Worst-1-DI Pareto curves for each method. Overall, ROAD mostly outperforms all other methods. This tends to show how our method efficiently maximizes local fairness, without sacrificing any other desirable criterion too much. On the other hand, BROAD does not always perform as effectively as ROAD, illustrating the benefit from the local smoothness induced by the use of a neural network. Interest-

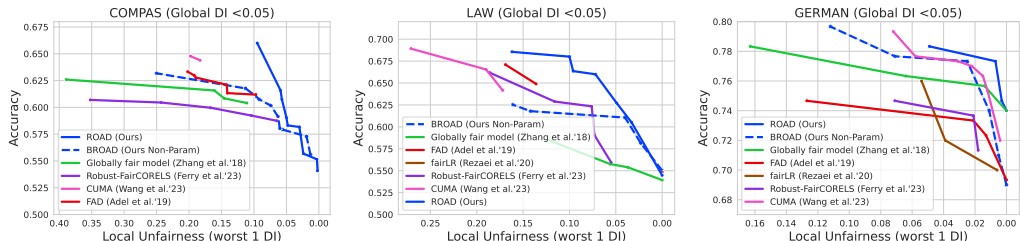

Figure 2: Results for the experiment on Local Fairness. For all datasets, the X-axis is Worst-1-DI, Y-axis is Global accuracy. The curves represented are, for each method, the Pareto front for the results satisfying the imposed global fairness constraint (here, Global DI < 0.05 for all datasets).

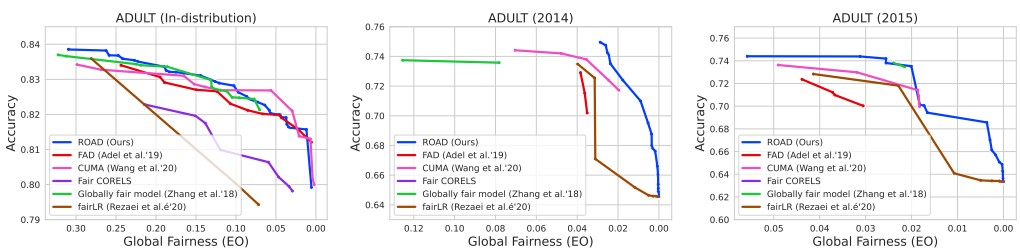

Figure 3: Pareto front results on distribution drift using the Adult dataset. For all figures, the X-axis is Equalized Odds; the Y-axis is Accuracy. Left: in-distribution (i.e. Adult UCI in 1994) test dataset; Center and Right: resp. 2014 and 2015 test datasets from Folktables (Ding et al., 2021).

ingly, despite not including any robustness component, globally fair methods of Zhang et al. (2018) and Adel et al. (2019) still manage to slightly reduce local bias through their global mechanisms.

## 4.2 EXPERIMENTS ON DISTRIBUTION DRIFT

As discussed in Section 2.3, DRO-based techniques have been considered before to help with the generalization of fairness. In this section, we therefore aim to show how our approach also leads to a better generalization of fairness in the face of distribution shift in addition to better-protecting subpopulations. For this purpose, we replicate the experimental protocol of Wang et al. (2023): after training classifiers on the training set of the classical Adult dataset (1994), we evaluate the tradeoff between accuracy and global fairness (measured with Equalized Odds (EO)) on the 2014 and 2015 Folktables datasets (Ding et al., 2021), containing US Census data from corresponding years, thus simulating real-world temporal drift. The same approaches as in the previous section, adapted to optimize for EO (details in Appendix A.8.2), are tested. Once again, the hyperparameters of every method are adjusted to maximize the two considered criteria, and the Pareto front is shown in Fig. 3.

Results on the classical Adult test set (in-distribution, left figure) are somewhat similar for most methods, with CUMA (Wang et al., 2023) slightly out-performing other methods. However, on drifted test sets (center and right figures), ROAD seems to achieve significantly better results than other methods, including other DRO-based fairness approaches. This suggests that the parametric implementation proposed in the paper is better suited to ensure robust behavior.

## 4.3 ABLATION STUDIES

### 4.3.1 BEHAVIOR OF $r$ AND IMPACT OF $\tau$

The behavior of ROAD depends on $\tau$, which controls the extent to which the distributions $q \in \mathcal{Q}$ are allowed to diverge from $p$. The impact of $\tau$ can be observed in the left figure of Fig. 4 for the Compas dataset. As values of $\tau$ increase, the variance of the distribution of $r$ decreases, going from having most weights close to $0$ and very high importance on a few others, to having most weights $r_i$ lying around $1$. Choosing the right value of $\tau$ thus helps control the emphasis put on some subpopulations.

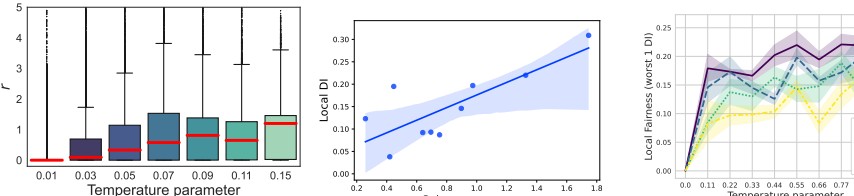

Figure 4: Analysis of the behavior of ROAD on Compas. Left: distribution of $r$ for several values of $\tau$ at epoch 200 (truncated at $r > 5$). Center: Relationship between Local DI and the average value of $r$ assigned to instances belonging to the corresponding subgroups. Each dot is a subgroup. Right: Worst-1-DI as a function of $\tau$ for different values for $\lambda_g$ (quartiles between $0.0$ and $10.0$)

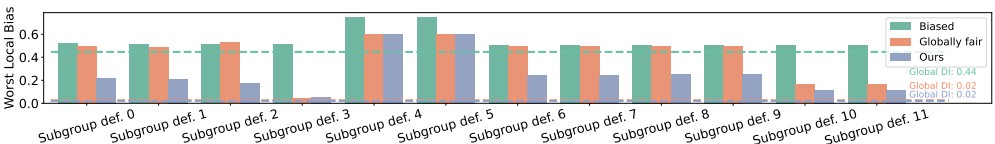

Figure 5: Worst-1-DI scores for subgroups of the Law dataset of various definitions, built by varying *age* bin width and splits along *gender*. Full description or the subgroups is available in Sec. A.8.3

A critical assumption ROAD relies on is that the adversary $r$ puts more attention on locally unfair regions. We test this assumption on the Compas dataset (same subgroups as in Sec. 4.1) and observe the results in the middle of Fig. 4. For each subgroup $k \in \mathcal{G}$ (blue dots), we measure its local fairness (y-axis) and the average weight $\mathbb{E}_{(x,s) \in k}(r_i(x,s))$ associated to instances of $k$. The graph reveals a correlation between these two notions, suggesting that more emphasis is indeed put on more unfair regions. As a consequence of these two results, setting $\tau$ helps control local bias, as shown in the right of Fig. 4 for various values of $\lambda_g$. The perfect local fairness score achieved when $\tau = 0$ is due to a constant model $f_{w_f}$: with no shape constraint, $r$ concentrates all the fairness effort on each training sample successively, which finally leads to $f(X) = \mathbb{E}[Y]$ for any input. Choosing a higher value of $\tau$ helps regularizing the process by inducing a distribution $q(x|s)$ closer to $p(x|s)$.

### 4.3.2 HOW IMPORTANT IS THE DEFINITION OF SUBGROUPS?

The main motivation for ROAD is its ability to maximize local fairness when the definition of the local subgroups is unknown. To assess the veracity of this claim, we conduct another experiment where we measure the local fairness of ROAD when the definition of these subgroups vary. Concretely, we train once a biased model, a globally fair model (Zhang et al., 2018) and ROAD (with resp. accuracy scores $0.72$, $0.60$, and $0.60$), and measure the local fairness for these models in subgroups of various definitions. These subgroups are defined successively as age bins with a width of 5, 10, 15 and 20, first across the whole population and then across subpopulations of other, non-sensitive, variables. Fig. 5 shows the local fairness results for the Law dataset (sensitive attribute is *Race*, subgroup attributes are *Age* and *Gender*). As expected, although the worst local DI for ROAD varies when the subgroup definition changes, it is almost consistently below the values reached by the globally fair model (except Def. 3 corresponding to the largest subgroups). This suggests that its tuning is not over-reliant on one subgroup definition, showcasing the flexibility of the approach.

## 5 CONCLUSION

In this work, we introduced the problem of enforcing local fairness in unknown subpopulations. By leveraging the strengths of adversarial learning and Distributionally Robust Optimization, our proposed framework ROAD provides a powerful approach for this setting, addressing the shortcomings of previous DRO-based approaches. Future works include extending our work to settings where the sensitive attribute is not available, to other differentiable penalties (e.g., Mutual Information in Ragonesi et al. (2021), and further exploring the optimization of a 3-network adversarial approach.

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

# A  APPENDIX

## A.1  DERIVATION OF ROAD (EQ. 13, WITH GLOBAL AND CONDITIONAL NORMALIZATION)

We start from the initial formulation of the problem as stated in equation 4, that we relax, and include the KL regularization to get the quantity $L_{\mathcal{Q}}$ to optimize:

$$L_{\mathcal{Q}} \triangleq \mathbb{E}_{p(x,y,s)}\left[\ \mathcal{L}_{\mathcal{Y}}(f_{w_f}(x), y)\right] - \lambda_g\left[\mathbb{E}_{q(x,s)}\left[\mathcal{L}_{\mathcal{S}}(g_{w_g*}(f_{w_f}(x)), s)\right] + \tau\mathbb{E}_{q(x,s)}\left[\log\frac{q(x,s)}{p(x,s)}\right]\right]$$

where $q \in \mathcal{Q}$, with $\mathcal{Q}$ the set of joint distributions over support of $(X, S)$ that are absolutely continuous with respect to $p(X, S)$.

Let us write $r(x, s) \triangleq \frac{q(x,s)}{p(x,s)}$. Thus, using the importance sampling trick, we get:

$$L_{\mathcal{Q}} \triangleq \mathbb{E}_{p(x,y,s)}\left[\ \mathcal{L}_{\mathcal{Y}}(f_{w_f}(x), y)\right] -$$
$$\lambda_g\left[\mathbb{E}_{p(x,s)}\left[r(x,s)\mathcal{L}_{\mathcal{S}}(g_{w_g*}(f_{w_f}(x)), s)\right] + \tau\mathbb{E}_{p(x,s)}\left[r(x,s)\log r(x,s)\right]\right]$$

Following this, considering a training dataset $\Gamma = (x_i, y_i, s_i)_{i=1}^n$ sampled from $p(X, Y, S)$, our optimization problem could be given as (which we refer to as single normalization):

$$\min_{w_f}\max_{r\in\mathcal{R}}\frac{1}{n}\sum_{i=1}^n\mathcal{L}_Y(f_{w_f}(x_i), y_i) - \lambda_g[\frac{1}{n}\sum_{i=1}^n r(x_i, s_i)\mathcal{L}_{\mathcal{S}}(g_{w_g^*}(f_{w_f}(x_i)), s_i)$$
$$+\tau\underbrace{\frac{1}{n}\sum_{i=1}^n r(x_i, s_i)\log(r(x_i, s_i))]}_{\text{KL constraint}}$$
$$\text{with } w_g^* = \arg\min_{w_g}\frac{1}{n}\sum_{i=1}^n\mathcal{L}_S(g_{w_g}(f_{w_f}(x_i)), s_i)$$

where $\mathcal{R} = \{r|pr \in \mathcal{Q}\}$ is an uncertainty set ensuring that $q = pr$ is a distribution (i.e., respecting $\mathbb{E}_p(x, s)[r(x, s)] = 1$).

As explained in section 3 (with further analysis in section A.2.2), this set is not restrictive enough to guarantee a stable optimization for our fairness objectives, for settings implying an adversarial trained on a dataset with unbalanced proportions of sensitive values (i.e., $p(S)$ is not uniform).

To go further, we propose to restrict to $\tilde{\mathcal{Q}} \subset \mathcal{Q}$, such that any $q \in \tilde{\mathcal{Q}}$ respects marginal equalities $q(s) = p(s)$ for any $s \in S$. This leads to:

$$L_{\tilde{\mathcal{Q}}} = \mathbb{E}_{p(x,y,s)}\left[\ \mathcal{L}_{\mathcal{Y}}(f_{w_f}(x), y)\right] -$$
$$\lambda_g\left[\mathbb{E}_{p(s)}\mathbb{E}_{q(x|s)}\left[\mathcal{L}_{\mathcal{S}}(g_{w_g*}(f_{w_f}(x)), s)\right] + \tau\mathbb{E}_{p(s)}\mathbb{E}_{q(x|s)}\left[\log\frac{q(x,s)}{p(x,s)}\right]\right]$$

From the continuity of distributions in $\mathcal{Q}$ w.r.t. $p$ and the fact that $q(s) = p(s)$ for any $q \in \tilde{\mathcal{Q}}$, $p(x|s) > 0$ whenever $q(x|s) > 0$. Thus, using the importance sampling trick and introducing $r(x|s) \triangleq \frac{q(x|s)}{p(x|s)}$, we get:

$$L_{\tilde{\mathcal{Q}}} = \mathbb{E}_{p(x,y,s)}\left[\ \mathcal{L}_{\mathcal{Y}}(f_{w_f}(x), y)\right] -$$
$$\lambda_g\left[\mathbb{E}_{p(s)}\mathbb{E}_{p(x|s)}\left[r(x|s)\mathcal{L}_{\mathcal{S}}(g_{w_g*}(f_{w_f}(x)), s)\right] + \tau\mathbb{E}_{p(s)}\mathbb{E}_{p(x|s)}\left[r(x|s)\log r(x|s)\right]\right]$$

Noting that for any $q \in \tilde{\mathcal{Q}}$, $r(x, s) = r(x|s)r(s) = r(x|s)$, this leads to the following optimization problem, with $\tilde{\mathcal{R}} = \{r|pr \in \tilde{\mathcal{Q}}\}$:

$$\min_{w_f} \max_{r \in \tilde{\mathcal{R}}} \frac{1}{n} \sum_{i=1}^{n} \mathcal{L}_Y(f_{w_f}(x_i), y_i) - \lambda_g [\frac{1}{n} \sum_{i=1}^{n} r(x_i, s_i) \mathcal{L}_S(g_{w_g^*}(f_{w_f}(x_i)), s_i)$$

$$+ \tau \underbrace{\frac{1}{n} \sum_{i=1}^{n} r(x_i, s_i) \log(r(x_i, s_i))]}_{\text{KL constraint}}$$

$$\text{with } w_g^* = \arg\min_{w_g} \frac{1}{n} \sum_{i=1}^{n} \mathcal{L}_S(g_{w_g}(f_{w_f}(x_i)), s_i)$$

which is equivalent to eq 13, since $\tilde{\mathcal{R}} = \{r \in \mathcal{R} | \forall s, \mathbb{E}_{p(x|s)} r(x, s) = 1\}$.

## A.2 WHY DOES USING TWO CONDITIONAL NORMALIZATION CONSTRAINTS INSTEAD OF A SINGLE GLOBAL ONE HELP?

### A.2.1 THEORETICAL IMPLICATION OF THE DOUBLE NORMALIZATION CONSTRAINT

This section aims at proving that using two conditional normalizations of the $r$ values, one for each sensitive, leads to guarantee $p(s) = q(s)$, as claimed in the main paper and advised in the next sections.

Assuming we have a function $r$ that respects the property: $\mathbb{E}_{p(x|s)} \ r(x, s) = 1$ for both $s$, we can first start by observing that the classical global constraint also holds: $\mathbb{E}_{p(x,s)} \ r(x, s) = \mathbb{E}_{p(s)} \mathbb{E}_{p(x|s)} \ r(x, s) = \sum_s p(s) = 1$. This allows us to consider $r(x, s) = \frac{q(x,s)}{p(x,s)}$ as a ratio of valid joint distributions, with $q(x, s)$ which integrates to one over the support of $p$.

Then, let us analyze the induced marginal $q(s)$ resulting from such implicit distribution:

$$q(s) = \int q(x, s) dx = \int p(x, s) \frac{q(x, s)}{p(x, s)} dx = p(s) \mathbb{E}_{p(x|s)} r(x, s) dx = p(s)$$

As a result, considering the uncertainty set $\tilde{\mathcal{R}} = \{r \in \mathcal{R} | \forall s, \mathbb{E}_{p(x|s)} \ r(x, s) = 1\}$ actually implies that the induced prior $q(s)$ equals the sensitive prior $p(s)$ observed in the dataset.

For completeness, we also show the reciprocal, that states that setting $r$ such that $p(s) = q(s)$ implies the considered constraints on $r$, which indicates that these validity constraints include all distributions $q \in \mathcal{Q}$ such that $q(s) = p(s)$. To do so, let us consider that $\forall s, p(s) = q(s)$. Thus, we have $p(s) = \int q(x, s) dx = \int p(x|s) \frac{q(x,s)}{p(x|s)} dx$. It directly follows that $\forall s, \int p(x|s) \frac{q(x,s)}{p(x|s)p(s)} dx = \mathbb{E}_{p(x|s)}[r(x, s)] = 1$.

Considering our validity constraints on $r$ is thus equivalent as setting $q(s) = p(s)$ for every $s$, if dealing with an explicit distribution $q$ was tractable.

### A.2.2 ANALYSIS OF NORMALIZED WEIGHTS FOR A FULLY FAIR MODEL

In the context of fairness, we argue that ensuring a classical global normalization constraint $\mathbb{E}_{p(x,s)} r(x, s) = 1$ during optimization is not sufficient to provide an accurate behavior with regards to the fair metric, as it may lead to concentrate the majority of the fairness effort to a specific sensitive subgroup.

To understand why, let us go back to the justification of using an adversarial for group fairness as introduced in Zhang et al. (2018), taking demographic parity as an illustrative example (while the same remains true for other objectives, such as equalized odds). The objective of Demographic Parity is to obtain a classifier that allocates equal chances to both subgroups of the population given the sensitive $S$. Formally, a classifier $f$ that respects: $P(\hat{Y} = 1 | S = 1) = P(\hat{Y} = 1 | S = 0)$, where we note $\hat{Y} = \hat{f}(X) = I(f(X) > 0.5)$. Equivalently in the binary case, we need to ensure $\mathbb{E}[\hat{Y} | S = 1] = \mathbb{E}[\hat{Y} | S = 0]$, or again $P(\hat{Y} = 1 | S = 1) = P(\hat{Y})$. During optimization, it is

difficult to build regularization terms on comparison of such quantities, both of them implying the classifier. Besides the non-differentiability induced by the use of an indicator function in the loss, it would require to backpropagate through estimators from both populations, which may either be subject to high variance or intractability. Rather, adversarial fairness Zhang et al. (2018) proposes to use an adversary network $g$ that attempts to reconstruct the sensitive $S$ from $f(X)$. The adversary is optimal when it outputs $\mathbb{E}[S|f(X)]$ accurately for any input, and the classifier is fully fair when $\mathbb{E}[S|f(X)] = P(S = 1|f(X)) = P(S = 1)$. In that case, we indeed have $P(S = 1, f(X)) = P(S = 1)P(f(X))$, which means independence and finally $P(\hat{f}(X)|S) = P(\hat{f}(X))$.

Now, let us assume a situation where the classifier is fully fair, which thus means that $P(S|f(X)) = P(S)$ for any $x \in X$. Let us also assume that we use an optimal adversary $g^*(f(X))$, that accurately outputs $P(S|f(X))$ for each $(X, S)$ given $f$. Let us look at the optimal values of $r$ given $f$ by using our non-parametric formulation BROAD (see section 3.2.1). While using a single global normalization constraint $\mathbb{E}_{p(x,s)} r(x, s) = 1$ in such a situation of a fully fair classifier, we have: $r_i \propto e^{-\mathcal{L}_S(g^*(f(x_i)), s_i))/\tau} = e^{\log(P(S=s_i))/\tau}$ for any sample $i$ of the dataset. Without loss of generality, let us consider that $P(S = 1) > P(S = 0)$. In that case, we thus have: $e^{\log(P(S=1))/\tau}/Z > e^{\log(P(S=0))/\tau}/Z$, for any constant $Z > 0$, including $Z = \frac{1}{n}\sum_{(x_i,s_i)\in\Gamma} e^{\log(P(S=s_i))/\tau}$. This means that in that situation, any sample $i$ such that $s_i = 1$ obtains a weight $r_i$ that is greater than the one for any sample $j$ such that $s_j = 0$. In other words, with a global normalization constraint, samples from the most populated demographic-sensitive group are more constrained than other ones. Improving accuracy will thus be easier for the least populated group, leading to a model with unbalanced error between the two populations.

On the other hand, when considering the conditional validity constraints, we get for that situation:

$$r_i = \frac{e^{\log(P(S=s_i))/\tau}}{\frac{1}{n_{s=s_i}}\sum_{j,s_j=s_i} e^{\log(P(S=s_i))/\tau}} = 1 \qquad \forall i \in \Gamma$$

which ensures a uniform fairness effort for every sample point from the dataset. In that ideal situation, this means that $q(x, s) = p(x, s)$, coming back to a classical adversarial fairness approach such as in Zhang et al. (2018).

### A.2.3  EMPIRICAL OBSERVATIONS UNDER PRACTICAL SETTINGS

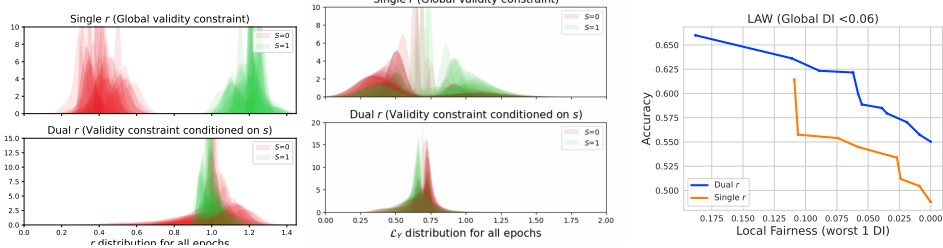

Figure 6: Effect of the conditional constraint $\mathbb{E}_{(x|s=1)} r(x, s) = \mathbb{E}_{(x|s=0)} r(x, s) = 1$. M Left: distributions of $r(x|s = 0)$ and $r(x|s = 1)$. Center: $\mathcal{L}_Y$ distributions. Right: Pareto fronts of the Accuracy-Local Fairness scores for the Law dataset.

In the previous section, we have shown that ROAD with a single global validity normalization over-constrains the most populated group regarding the sensitive attribute, in the case of a fully fair classifier. We claim that this analysis of an extreme setting highlights a general behavior of the method, which better balances fairness constraints when using our conditional validity normalization.

In this section, we propose to verify this assumption empirically, by considering the Law dataset, where the sensitive distribution is heavily imbalanced ($P(S = 1) \approx (1/4)P(S = 0)$). In the left image of Fig. 6 is shown the distribution of $r$ values assigned to instances verifying $S = 1$ (orange) and $S = 0$ (green) for various training iterations, using a global normalization in the top-most graph, and the conditional normalization on the bottom-most one. Using the global normalization, we observe that no overlap ever happens between the $r$ values of these two sensitive groups. As

a result, most part of the fairness effort is supported by individuals from the largest population $S = 0$. This is not the case with our proposed conditional normalization, which results in better local fairness (right image of fig. 6). We also observe in the center figure that the mitigation effort is greatly better balanced when using our conditional validity constraint.

While our conditional validity constraint that implies $q(s) = p(s)$ may look as limitative in case of distributional drifts (notably implying a drift of $p(s)$), we claim this is not the case as it does not constrain distributions inside both sensitive groups. First, we want to point out that distribution drift on $p(s)$ without change on the support (i.e., with a drifted distribution $p_{test}$ that respects the absolute continuity hypothesis of $q$ w.r.t. to $p_{test}$) does not have any impact on the fairness of a fully locally fair model. It may only impact accuracy, which could be dealt with a second reweighting scheme for the prediction loss, same manner as done in classical DRO works such as Michel et al. (2022) (this was not considered in our work to avoid over-complexity of presentation and highlight our innovation, but we acknowledge the potential interest and value of this complementary direction).

In case of models where subgroups do not ensure $P(\hat{Y}|S) = P(\hat{Y})$, distributional drift may exhibit these local mistreatments with more emphasis on them in the test distribution, which explains the effectiveness of our approach for such setting (drift may make some subgroups more represented in the global evaluation metrics). On the other hand, drifting p(s), which corresponds to a global drift, is however not supposed to do so if the fairness effort is well balanced over both sensitive groups (as it is shown to be in figure 6 when using our conditional validity constraint). This is verified in our experiments corresponding to figure 3, for which a drift on p(s) is applied (from $66\%$ male in train, for $53\%$ in 2014 and 2015 test sets), without limiting the effectiveness of our approach.

## A.3 THEORETICAL PROOF: BROAD IS A BOLTZMANN DISTRIBUTION

This section contains proof for Lemma 3.1, that we rewrite below:

**Lemma A.1.** *(Optimal Non-parametric Ratio) Consider the optimization problem (inner maximization problem, on $w_r$, of Equation 13):*

$$\max_{w_r} \mathbb{E}_{(x,y,s)\sim p}(r(x,s)(\mathcal{L}_S(g_{w_g^*}(f_{w_f}(x)),s)) + \tau \mathbb{E}_{(x,y,s)\sim p}(r(x,s)\log r(x,s) \tag{7}$$

$$\text{with } w_g^* = \arg\min_{w_g} \mathbb{E}_{(x,y,s)\sim p}\mathcal{L}_S(g_{w_g}(f_{w_f}(x)),s)$$

$$\text{Under the global validity constraint: } \mathbb{E}_{(x,y,s)\sim p}\, r(x,s) = 1$$

*The solution to this problem can be rewritten as an inverted Boltzmann distribution with a multiplying factor $n$:*

$$r(x_i,s_i) = \frac{e^{-\mathcal{L}_S(g_{w_g}(f_{w_f}(x_i)),s_i)/\tau}}{\frac{1}{n}\sum_{j=1}^{n} e^{-\mathcal{L}_S(g_{w_g}(f_{w_f}(x_j)),s_j)/\tau}}$$

*Proof.* The proof results from a direct application of the Karush-Kuhn-Tucker conditions. Since there is no ambiguity, we use the following lighter notation for the sake of simplicity: $r$ instead of $r(x,s)$; $\mathcal{L}_S$ instead of $\mathcal{L}_S(g_{w_g^*}(f_{w_f}(x)),s))$; and $\mathbb{E}$ instead of $\mathbb{E}_{(x,y,s)\sim p}$. Using these notations and writing the above problem in its relaxed form w.r.t. the distribution constraint, we obtain the following formulation:

$$\max_{r} \mathbb{E}(r\mathcal{L}_S) + \tau \mathbb{E}(r\log r) - \kappa(1 - \mathbb{E}(r))$$

Following the Karush-Kuhn-Tucker conditions applied to the derivative of the Lagrangian function $L$ of this problem in $r_i$, we obtain:

$$\frac{\partial L}{\partial r_i} = 0 \Leftrightarrow \mathcal{L}_S^i + \tau(1 + \log r_i) + \kappa = 0 \Leftrightarrow r_i = e^{\frac{-\kappa - \mathcal{L}_S^i}{\tau} - 1} \tag{8}$$

With $\mathcal{L}_S^i = \mathcal{L}_S(g_{w_g^*}(f_{w_f}(x_i)),s_i)$ the i-th component of $\mathcal{L}_S$. The KKT condition on the derivative in $\kappa$ gives: $\frac{\partial L}{\partial \kappa} = 0 \Leftrightarrow \mathbb{E}(r) = 1$. Combining these two results, we thus obtain:

$$\mathbb{E}(r) = \frac{1}{n}\sum_{j=1}^{n} r_j = \frac{1}{n}\sum_{j=1}^{n} e^{\frac{-\kappa - \mathcal{L}_S^j}{\tau} - 1} = 1 \Leftrightarrow e^{-\frac{\kappa}{\tau} - 1} = \frac{1}{\frac{1}{n}\sum_{j=1}^{n} e^{-\frac{\mathcal{L}_S^j}{\tau}}}$$

Which again gives, reinjecting this result in Eq. 8:

$$r_i = \frac{\mathrm{e}^{-\frac{\mathcal{L}_S^i}{\tau}}}{\frac{1}{n}\sum_{j=1}^{n}\mathrm{e}^{-\frac{\mathcal{L}_S^j}{\tau}}}$$

This leads to the form of a Boltzmann distribution (ignoring a multiplication factor $n$), which proves the result. $\qquad\square$

The same proof can be derived under the conditional validity constraint.

### A.4    ADDITIONAL ABLATION STUDIES

In this section, we present several additional results, that were not included in the paper due to lack of space.

#### A.4.1    DOES THE COMPLEXITY OF THE TWO ADVERSARIAL NETWORKS HAVE AN IMPACT ON THE QUALITY OF THE RESULTS?

In this section, we investigate the impact of the complexity of the two adversarial networks $g$ and $r$ on the quality of the results. Previous works, such as Grari et al. (2019), have studied the influence of the adversary $g$ complexity on the context of fair adversarial learning objectives and show that a complex architecture can help to achieve significantly better results than a simple one (i.e., a logistic regression).

Figure 7 presents the results of this study on the LAW dataset. We compare three levels of complexity for each network: simple (linear predictor), medium (only one hidden layer with 32 neurons with a ReLu activation function), and complex (three hidden layers: 64 neurons, ReLu, 32 neurons, Relu, 16 neurons, Relu). The results show that a more complex adversary $g$ tends to yield better results, which is consistent with the observations made by Grari et al. (2019) (i.e., a complex network achieves better performance than a simple logistic regression). One possible reason is that a single logistic regression may not retain enough information to predict the sensitive attribute accurately.

In contrast, for network $r$, a simpler linear prediction such as logistic regression is quite efficient (please note that due to the exponential parametrization, an exponential activation is applied to it), while a more complex architecture results in lower performance. As explained in Section 3.2.2, the network architecture plays a crucial role in determining the local behavior of $r_{w_r}$. More complex networks tend to favor more local solutions for a given value of $\tau$. In particular, a network of infinite capacity that completes the training will, in theory, exhibit the same behavior as BROAD, thereby yielding more pessimistic solutions.

This underscores the importance of carefully considering the complexities of the adversaries when designing a fair adversarial learning framework. Our analysis suggests that a more complex adversary $g$ is preferable, while a simpler network $r$ can be more efficient. Further research is needed to better understand the interplay between these complexities and develop strategies for selecting the optimal combination for a given problem setting.

#### A.4.2    ASSESSING LOCAL FAIRNESS ON THE WORST-3-DI INSTEAD OF THE WORST-1-DI

In this experiment, we aim to gain a deeper understanding of our methods' behavior. Instead of assessing local fairness based on the worst disparate impact value for the globally defined subgroups, we focus on the worst 3. This approach allows us to observe the impact of our instance-level re-weighting strategy at an intermediate level.

The definition of our segment remains consistent as in Section 4.1. In Figure 8 we show the resulting Accuracy-Worst-3-DI Pareto curves for each method. We observe that the results are slightly similar to those observed on the Worst-1-DI in the main text. For all datasets, as in the Worst-1-DI experiment ROAD mostly outperforms all other methods. On the other hand, BROAD sometimes reaches comparable performance, illustrating the overly pessimistic solutions mentioned earlier.

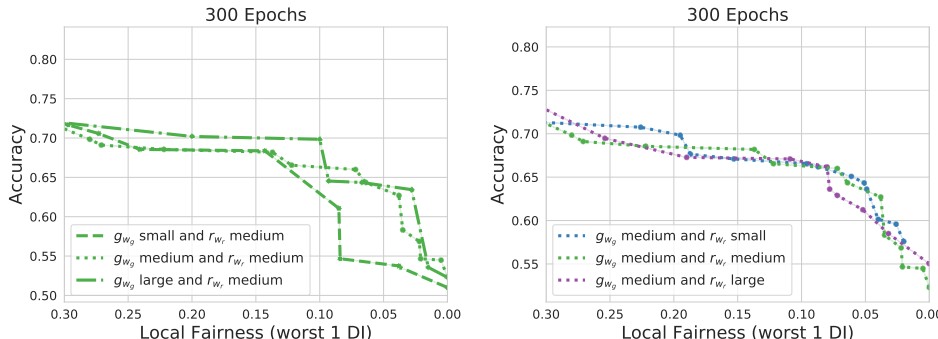

Figure 7: Impact of the complexity of adversarial networks g and r on the quality of results on the LAW dataset. The comparison is carried on three complexity levels: simple (a simple regression), medium (only one hidden layer), and complex (three hidden layers). The curves represented are, for each method, the Pareto front for the results satisfying the imposed global fairness constraint (here, Global DI < 0.05 for all datasets).

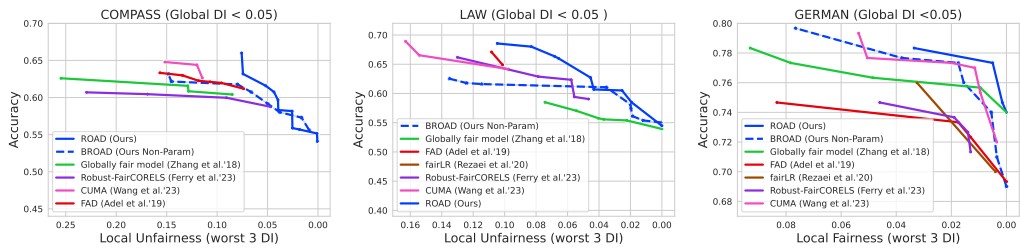

Figure 8: Results for the experiment on Local Fairness. For all datasets, the X-axis is Worst-3-DI, Y-axis is Global accuracy. The curves represented are, for each method, the Pareto front for the results satisfying the imposed global fairness constraint (here, Global DI < 0.05 for all datasets).

### A.4.3 ADDITIONAL RESULTS FOR THE ABLATION STUDY ON SUBGROUP DEFINITION (SECTION 4.3.2)

Section 4.3.2 described the experimental results on the adaptability of ROAD to various subgroup definitions. We complete these results with the following: in Figure A.4.3, the experiments are repeated five times, and the same conclusions are observed: It is consistently below the values reached by the globally fair model Zhang et al. (2018) In addition; in Figure A.4.3, the details of the same results are presented. Instead of only showing the Worst-1-DI values, all local DI values, for every subgroup, are shown.

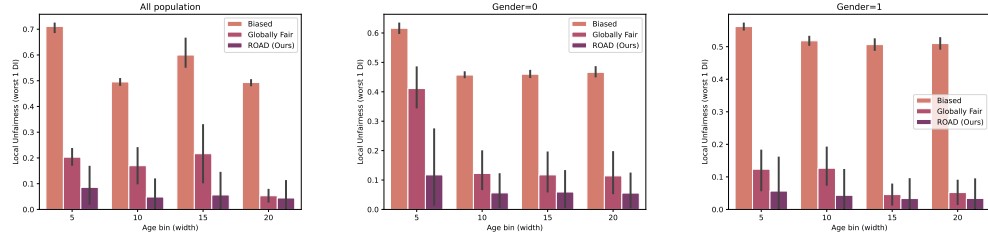

Figure 9: Local DI scores for subgroups of the Law dataset defined as age bins (top row) and age bins and gender attributes (middle and bottom rows). Each column corresponds to bins of different sizes. Subgroups with less than 10 individuals are ignored. The process has been iterated five times.

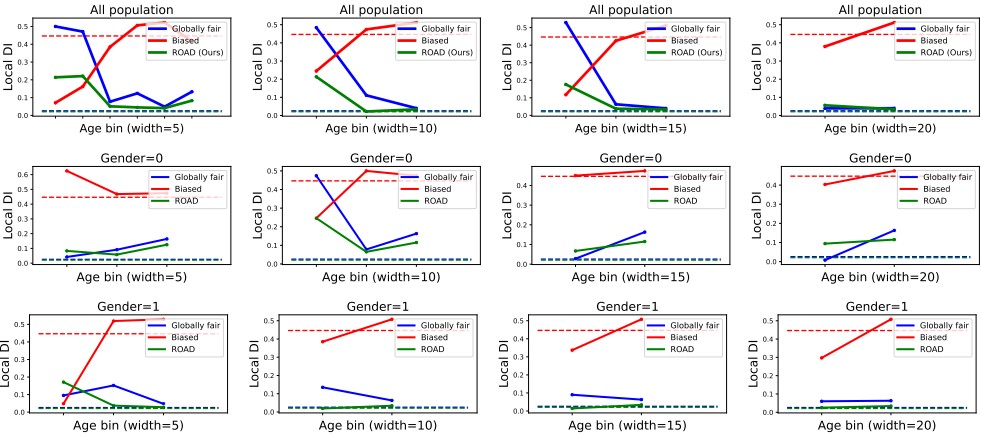

Figure 10: Local DI scores for subgroups of the Law dataset defined as age bins (top row) and age bins and gender attribute (middle and bottom rows). Each column corresponds to bins of different sizes. Subgroups with less than 10 individuals are ignored.

## A.5 ADDITIONAL RESULTS ON THE DIFFERENCE WITH TRADITIONAL PARAMETRIC DRO

In this section, we provide additional results and discussion to further clarify the differences between our approach and parametric Distributionally Robust Optimization (DRO) approach such as presented in Michel et al. (2022). The main distinctions can be summarized as follows:

1. **Different problem formulation and objective:** Our method focuses on optimizing fairness criteria, such as Demographic Parity, by considering a sensitive attribute $S$. In contrast, Michel et al. (2022) apply parametric DRO to the training objective $L_Y$ without considering any fairness criterion. This fundamental difference in problem formulation leads to distinct optimization objectives, with our approach.

2. **Different optimization:** Our approach maximizes the minimum (max-min)possible adversarial loss and assigns more weight to samples that are easier for the adversary to reconstruct $S$. This is in stark contrast with the traditional DRO pursued by Michel et al. (2022), which minimizes the maximum (min-max) training loss and focuses on classifier accuracy without considering fairness. Our technique employs a multi-objective optimization approach that balances fairness and accuracy, while Michel et al. (2022)prioritize optimizing accuracy.

3. **Different neural network architectures:** Our optimization process involves three neural networks, while Michel et al. (2022) utilize two. This difference in architecture complexity is a direct result of addressing distinct problem formulations and optimization goals in each work.

4. **Different validity constraint:** The validity constraints used in our work and Michel et al. (2022) differ due to the unique optimization objectives and problem formulations. Our approach focuses on satisfying a fairness constraint while optimizing performance, whereas Michel et al. (2022) primarily concentrate on optimizing accuracy.

In Figure 11, we present the results for the experiment on Local Fairness for the German dataset. The X-axis represents the Worst-1-DI, while the Y-axis corresponds to Global accuracy. We compare the Pareto front of our method (DRO) with that of Michel et al. (2022). From the figure, we observe that our method consistently achieves better results in the Pareto front compared to Michel et al. (2022). Our approach demonstrates a more favorable trade-off between Worst-1-DI and Global accuracy, indicating that it can achieve higher accuracy while maintaining a higher degree of fairness (lower Worst-1-DI values) than the method proposed by Michel et al. (2022).

This result further supports the novelty and effectiveness of our approach, as it not only addresses a distinct problem formulation and employs a unique optimization strategy, but also demonstrates

improved performance in terms of both fairness and accuracy compared to existing methods, such as Michel et al. (2022).

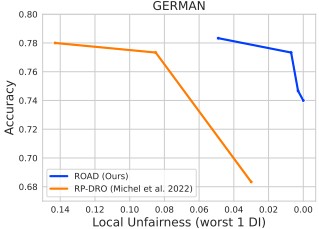

Figure 11: Results for the experiment on Local Fairness. For all datasets, the X-axis is Worst-1-DI, Y-axis is Global accuracy. The curves represented are, for each method, the Pareto front for the results satisfying the imposed global fairness constraint (here, Global DI < 0.05 for all datasets).

### A.6 ADDITIONAL RESULTS ON THE LOCAL EQUALIZED-ODDS CRITERION

As mentioned in the section2 **Equalized Odds** is another group fairness criterion. A classifier is considered fair according to this criterion if the outcome $\hat{f}_{w_f}(x)$ has equal *false* positive rates and *false* negative rates for both demographics $S = 0$ and $S = 1$ Hardt et al. (2016). This criterion is especially relevant when misclassification can have significant impacts on individuals from different groups. A metric to assess this is the disparate mistreatment (DM) Zafar et al. (2015), which we report as the sum of the two following quantities:

$$\Delta_{FPR} : |P(\hat{f}_{w_f}(x) = 1|y = 0, s = 1) - P(\hat{f}_{w_f}(x) = 1|y = 0, s = 0)|$$
$$\Delta_{FNR} : |P(\hat{f}_{w_f}(x) = 0|y = 1, s = 1) - P(\hat{f}_{w_f}(x) = 0|y = 1, s = 0)|$$

As in experiment 4, to assess equalized odds at a local level, we have chosen various subpopulations among features of $X$, i.e. excluding $S$ selected in the test set. As measuring fairness criteria in segments of low population is highly volatile, we filter out subgroups containing less than $50$ individuals (more details in Appendix A.8.3). These subgroups are unknown at training time, and chosen arbitrarily to reflect possible important demographic subgroups (see Section 4.3.2 for further discussion). Given the set of these subgroups $\mathcal{G}$, the local fairness is then assessed on the worst False Positive Rate among these subgroups: $Worst\text{-}1\text{-}\Delta_{FPR} = \max_{g \in \mathcal{G}} |\mathbb{E}_{(x,s) \in g}(\hat{f}_{w_f}(x)|s = 1, y = 0) - \mathbb{E}_{(x,s) \in g}(\hat{f}_{w_f}(x)|s = 0, y = 0)|$, on the False Negative Rate among these subgroups: $Worst\text{-}1\text{-}\Delta_{FPR} = \max_{g \in \mathcal{G}} |\mathbb{E}_{(x,s) \in g}(\hat{f}_{w_f}(x)|s = 1, y = 0) - \mathbb{E}_{(x,s) \in g}(\hat{f}_{w_f}(x)|s = 0, y = 0)|$ and the worst Disparate Mistreatment Rate among these subgroups: $Worst\text{-}1\text{-}\Delta_{DMR} = \max_{g \in \mathcal{G}} |\mathbb{E}_{(x,s) \in g}(\hat{f}_{w_f}(x)|s = 1, y = 0) - \mathbb{E}_{(x,s) \in g}(\hat{f}_{w_f}(x)|s = 0, y = 0)| + \mathbb{E}_{(x,s) \in g}(\hat{f}_{w_f}(x)|s = 1, y = 0) - \mathbb{E}_{(x,s) \in g}(\hat{f}_{w_f}(x)|s = 0, y = 0)|$

To evaluate our approach, we compare our results with adversarial models specifically adapted for equalized odds from Zhang et al. (2018) and Adel et al. (2019), as well as 3 existing works that aim to address fairness generalization: FairLR (Rezaei et al., 2020), and CUMA (Wang et al., 2023) (cf. Appendix A.8.2 for more details). While we keep the same name there are different than demographic parity, for example, to achieve EO, Zhang et al. (2018) employs an adversarial learning approach by concatenating the outcome prediction $f(x)$ the true outcome $Y$ to the input of the adversary. The resulting relaxed optimization problem can be expressed as $\min_{w_f} \max_{w_g} \mathbb{E}_p \mathcal{L}_{\mathcal{Y}}(f(x), y) - \lambda \mathbb{E}_p \mathcal{L}_{\mathcal{S}}(g_{w_g}(f_{w_f}(x), y), s)$. This methodology balances minimizing the predictor model's loss and maximizing the adversarial model's loss while satisfying equalized odds.

As local fairness can only be measured against global accuracy and global fairness, we evaluate the approaches by plotting the tradeoffs between global accuracy, $Worst\text{-}1\text{-}\Delta_{FPR}$, $Worst\text{-}1\text{-}\Delta_{FNR}$ and $Worst\text{-}1\text{-}\Delta_{DMR}$ subject to a global FPR and FNR constraint (we choose a threshold equal 0.05, a common threshold in the fairness literature (Pannekoek & Spigler, 2021)). As in the section 4 to ensure a thorough exploration of these tradeoffs, we sweep across hyperparameter values for each

algorithm (details of hyperparameter grids in App. A.8.4). Figure 2 shows the resulting Accuracy-*Worst-1-*$\Delta_{DMR}$, Accuracy- *Worst-1-*$\Delta_{FNR}$ and the Accuracy- *Worst-1-*$\Delta_{FPR}$ Pareto curves for each method respectively.

For all datasets, ROAD mostly outperforms all other methods. This tends to show how the proposed method efficiently maximizes local fairness, without sacrificing any other desirable criterion too much. On the other hand, BROAD does not always perform as effectively as ROAD, illustrating the benefit resulting from the local smoothness induced by the use of a neural network.

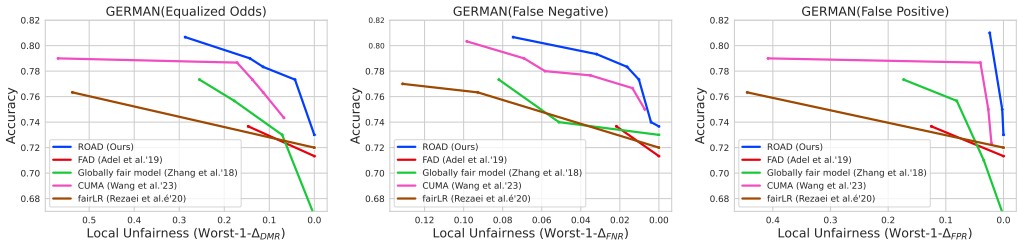

Figure 12: Results for the experiment on Local Fairness. For all datasets, Y-axis is Global accuracy. The curves represented are, for each method, the Pareto front for the results satisfying the imposed global fairness constraint (here, Global FNR and Global FPR $< 0.05$ for all datasets).

## A.7 ADDITIONAL RESULTS ON THE CONDITIONED VALIDITY CONSTRAINT

In this experiment, our objective is to empirically compare two validity constraints: one considering only the conditionality with $S$, as previously discussed, and another that incorporates, in addition, the true value of the $y$ outcome in the expectation of the ratio. This approach stems from the idea that the mitigation of the disparate mistreatment A.6 should be divided into the two different values of $Y$ to account for both false positives and false negatives per sensitive groups.

The conditional validity constraint of interests takes the following form:

$$\forall s_i, \forall y_i, \mathbb{E}_{x|s=s_i,y=y_i} r(x,s,y) = 1$$

For each sample $(x_i, y_i, s_i)$ in the mini-batch, we define the normalized ratio as:

$$\forall i, \ r_{w_r}(x_i, s_i, y_i) = \frac{e^{h_{w_r}(x_i,s_i,y_i)}}{\frac{1}{n_{s_i,y_i}} \sum_{(x_j,s_j,y_j)\in\Gamma, s_j=s_i, y_j=y_i} e^{h_{w_r}(x_j,s_j,y_j)}}$$

Here, as before $h : \mathcal{X} \times 0; 1 \rightarrow \mathcal{R}$ is a neural network with weights $w_r$.

As previously mentioned, we employ an exponential parametrization with two batch-level normalizations, one for each demographic group. Moreover, due to the conditionality with $Y$, we incorporate two additional normalizations, leading to a total of four demographic groups of interest.

To test the efficacy of this approach, we replicate the experimental protocol of the drift experiment described in subsection 4.2. Results on the classical validity constraint (validity constraint on $S$) and with the addition of $Y$ (validity constraint under $Y$ and $S$) are presented in Figure 13. Surprisingly, the results show that the validity constraint with only $S$ leads to a better generalization of fairness in the face of distribution shift. One assumption for interpreting these results is that the decomposition in four groups, i.e. along both $Y$ and $S$, may lead to very small in-batch subsamples, i.e. non-significant expectation estimates.

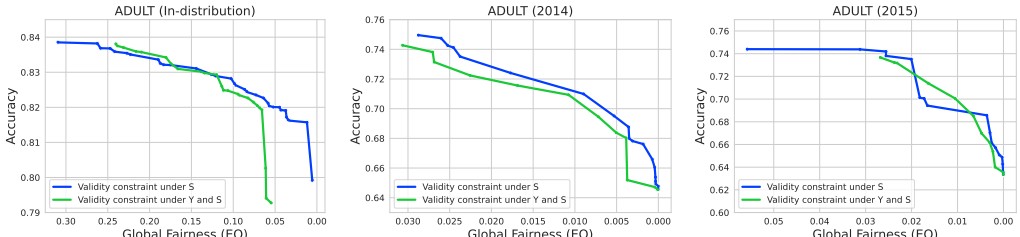

Figure 13: Pareto front results on distribution drift using the Adult dataset. For all figures, the X-axis is Equalized Odds; the Y-axis is Accuracy. Left: in-distribution (i.e. Adult UCI in 1994) test dataset; Center and Right: resp. 2014 and 2015 test datasets from Folktables (Ding et al., 2021).

## A.8    ADDITIONAL EXPERIMENTAL DETAILS

We provide additional details on the experimental evaluations.

### A.8.1    DATASETS DESCRIPTION

**Experiment 1: Assessing Local Fairness**

- **Compas** The COMPAS data set Angwin et al. (2016) contains 13 attributes of about 7,000 convicted criminals with class labels that state whether or not the individual recidivated within 2 years. Here, we use race as sensitive attribute, encoded as a binary attribute, Caucasian or not-Caucasian.

- **Law** The Law School Admission dataset Wightman (1998) contains 10 features on 1,823 law school applicants, including LSAT scores, undergraduate GPA, and class labels representing their success in passing the bar exam. The sensitive attribute selected in this dataset is race, a binary attribute.

- **German** The German UCI credit dataset Hofmann (1994) comprises credit-related data for 1,000 individuals, including 20 attributes such as credit history, purpose, and employment status, as well as class labels that represent their credit risk (good or bad). For the sensitive attribute, we consider the gender attribute with Males as the privileged group, and Females as the unprivileged group.

**Experiment 2: Distribution Drift**    The training set used for this experiment is the training set of the traditional Adult dataset.

The test sets are the following:

- in-distribution: traditional test set of the Adult dataset
- 2014: Adult 2014 dataset from Folktables, see below
- 2015: Adult 2015 dataset from Folktables, see below

Adult dataset contains US Census data collected before 1996. Its very frequent use in the fairness literature was discussed in Ding et al. (2021), in which the authors criticize its out-of-date aspect, especially in the context of evaluating social aspects of algorithms such as their fairness. Instead, they introduce the Folktables, up-to-date, aligned, versions of the Adult dataset. Using these datasets, the authors discussed the generalization of the fairness metrics over time. A similar setup was then considered by Wang et al. (2023) to evaluate how robust their method, CUMA, was to temporal drift. In our work, we directly replicate their experimental setup.

### A.8.2    COMPETITORS AND BASELINES

Besides ROAD and BROAD, proposed in this work, the following method are used as competitors and baselines for the experiments:

- Globally fair model (Zhang et al., 2018): described in Section 2.1. To obtain the Pareto curves in the results, the $\lambda$ hyperparameter is set to various values, allowing the exploration of the Fairness-accuracy tradeoff. As described in the footnote in Section 3, this approach can be adapted to optimize either Demographic Parity or Equalized Odds.

- FAD (Adel et al., 2019): similarily to Zhang et al. (2018), FAD relies on an adversarial network to reconstruct the sensitive attribute and thus estimate bias. However, instead of reconstructing the sensitive attribute from the predictions $f(x)$, FAD reconstructs it from an intermediary latent space. The authors have shown that this helped obtaining better results. Similarly to Zhang et al. (2018), FAD relies on a single hyperparameter $\lambda$ to balance between accuracy and fairness. Originally optimizing Demographic Parity, the same adaption as the approach from Zhang et al. (2018) can be made to optimize EO. Although we have this method, it must be noted that several other notable adversarial learning for fairness approaches exist (Louppe et al., 2017; Wadsworth et al., 2018; Grari et al., 2022),

- Robust FairCORELS (Ferry et al., 2022): this method leverages the DRO framework for fairness and integrates in the FairCORELS approach (Aïvodji et al., 2021). It relies on preprocessing step discretizing the whole dataset into segments. The uncertainty set $\mathcal{Q}$ for DRO is then defined as a Jaccard Index-ball around $p$. RobustFairCORELS then uses linear programming and efficient pruning strategies to explore $\mathcal{Q}$ and solve its otpimization problem. RobustFairCORELS relies mainly on one hyperparameter: $\epsilon$, the unfairness tolerance allowed, from which is derived the *size* of $\mathcal{Q}$ (maximum Jaccard distance from $p$ allowed). Besides, it relies on FairCORELS hyperparameters, which we set to their default values. RobustFairCORELS can be used to optimize several fairness criteria, including Demographic Parity and Equalized Odds.

- fairLR (Rezaei et al., 2020): to achieve more robust fairness, fairLR also leverages the DRO framework for fairness. The generalization they propose to ensure is more restrictive, as they make the assumption that there is no drift between the train and test distributions. They therefore define the ambiguity set $\mathcal{Q}$ as the set of distributions matching some statistics of the training data, that they use to train logistic regressions. These constraints can be adapted either to optimize for Demographic Parity or Equalized Odds. fairLR relies only one hyperparameter, $C$, a regularization parameter for the logistic regression. Although it has seemingly no direct link with fairness, we still consider it in the experiments.

- CUMA (Wang et al., 2023): this method leverages adversarial learning to mitigate fairness. Besides, it proposes to add a new component to its loss function, based on the curvature smoothness of its loss function. The idea is that to ensure better fairness generalization, the curvatures of the loss functions of each sensitive group should be similar. The objective function is then approximated using a neural network. It uses two hyperparameters, to balance accuracy with fairness and smoothness: $\alpha$ (adversarial component) and $\gamma$ (smoothness component). Similarly to the Globally Fair model and FAD, CUMA can be adapted in the same manner to either optimize Demographic Parity of Equalized Odds.

### A.8.3 SUBGROUPS DESCRIPTION

In this section, we describe how subgroups were defined in the experiments.

**Experiment 1**

- **Compas** Subpopulations are created by discretizing the feature *Age* in buckets with a width of 10, intersected with feature *Gender*. Subgroups of size of at least 50 are kept.

- **Law** Subpopulations are created in the same manner, by discretizing the feature *Age* in buckets with a width of 10, intersected with feature *Gender*. Subgroups of size of at least 20 are kept.

- **German** Subpopulations are created by discretizing the feature *Age* in buckets with a width of 10, intersected with feature *Gender*. Subgroups of size of at least 20 are kept.

**Ablation study: how important is the definition of subgroups** Table 1 describes how subgroups were defined in the 12 scenarios shown in Figure 10.

| Subgroup definition | Age bin width | Gender |
|---|---|---|
| Subgroup def. 0 | 5 | All population |
| Subgroup def. 1 | 10 | All population |
| Subgroup def. 2 | 15 | All population |
| Subgroup def. 3 | 20 | All population |
| Subgroup def. 4 | 5 | Gender=0 |
| Subgroup def. 5 | 10 | Gender=0 |
| Subgroup def. 6 | 15 | Gender=0 |
| Subgroup def. 7 | 20 | Gender=0 |
| Subgroup def. 8 | 5 | Gender=1 |
| Subgroup def. 9 | 10 | Gender=1 |
| Subgroup def. 10 | 15 | Gender=1 |
| Subgroup def. 11 | 20 | Gender=1 |

Table 1: Description of how subgroups were defined in the 12 scenarios shown in Figure 5.

### A.8.4 IMPLEMENTATION DETAILS FOR ROAD AND BROAD

In order to obtain the results shown in Figure 2 ad 3, we explore the following hyperparameter values for ROAD and BROAD:

- $\lambda_g$: grid of 20 values between 0 and 5
- $\tau$: grid of 10 values between 0 and 1

The networks $f_{w_f}$, $g_{w_g}$ and $r_{w_r}$ have the following architecture:

- $f_{w_f}$: FC:64 R, FC:32 R, FC:1 Sig
- $g_{w_g}$: FC:64 R, FC:32 R, FC:16 R, FC:1 Sigm
- $h_{w_r}$: FC:64 R, FC:32 R, FC:1

### A.9 DESCRIPTION OF THE ALGORITHMS

### A.9.1 ROAD ALGORITHM

The ROAD algorithm, for the Demographic Parity objective, aims to train a locally fair model by iteratively updating three components: the sensitive adversarial model $g_{w_g}$, the adversarial ratio model $r_{w_r}$, and the predictor model $f_{w_f}$. The following problem of ROAD is defined as follows (cf. Eq. 4):

$$\min_{w_f} \max_{w_r} \frac{1}{n} \sum_{i=1}^{n} \mathcal{L}_Y(f_{w_f}(x_i), y_i) - \lambda_g \Big[ \frac{1}{n} \sum_{i=1}^{n} (\tilde{r}_{w_r}(x_i, s_i) \mathcal{L}_S(g_{w_g^*}(f_{w_f}(x_i)), s_i)))$$

$$+ \tau \underbrace{\frac{1}{n} \sum_{i=1}^{n} (\tilde{r}_{w_r}(x_i, s_i) \log(\tilde{r}_{w_r}(x_i, s_i)))}_{\text{KL constraint}} \Big] \tag{9}$$

$$\text{with } w_g^* = \arg\min_{w_g} \frac{1}{n} \sum_{i=1}^{n} \mathcal{L}_S(g_{w_g}(f_{w_f}(x_i)), s_i)$$

with $\tilde{r}$ representing the normalized ratio per demographic sub-groups.

To obtain better results, as commonly done in the literature, we perform several training iterations of the adversarial networks for each prediction iteration. This approach results in more robust adversarial algorithms, preventing the predictor classifier from dominating the adversaries.

The required inputs for ROAD include the training data containing $X$, $Y$, and $S$ (please note that $S$ is not required at the testing time), the loss functions (i.e., logloss function applied in our experiment), batch size, number of epochs, neural networks architectures, learning rates, and the control parameters ($\tau$ and $\lambda$).

For each epoch, the algorithm 1 iterates through all the batches in the training data. In each batch, the algorithm performs the following steps:

- Update the sensitive adversarial model $g_{w_g}$: The sensitive adversarial takes the current output predictions $f_{w_f}(x)$ as input. The sensitive adversarial loss is calculated based on the current parameters of the predictor and sensitive adversarial models. The sensitive adversarial model is then updated using traditional gradient descent.

- Update the ratio model $r_{w_r}$: The likelihood ratio is normalized across demographic subgroups. The likelihood ratio cost function is calculated with respect to the average of the product between the weighting ratio and sensitive adversarial loss in addition to the KL constraint. Then, the adversarial model is updated using gradient descent.

- Update the predictor model: The loss function for the predictor model $f_{w_f}$ is calculated, taking into account the traditional loss function of the predictor and the average of the product between the weighting ratio and sensitive adversarial. The predictor model is then updated using gradient descent.

The process is repeated for the specified number of epochs. By iteratively updating these three models, the algorithm aims to achieve a balance between minimizing the loss function and maintaining fairness across demographic subgroups in the resulting predictor model.

---

**Algorithm 1** ROAD: Robust Optimization for Adversarial Debiasing

---

**Require:** Training set $\Gamma$,
Individual loss functions $l_Y$ and $l_S$,
Batchsize $b$, Number of epochs $n_e$, Neural Networks $r_{\omega_r}$, $f_{\omega_f}$ and $g_{\omega_g}$,
Learning rates $\alpha_f$, $\alpha_g$ and $\alpha_r$, Number of training iterations $n_g$ and $n_r$,
Fairness control $\lambda_g \in \mathbb{R}$ and Temperature control $\tau \in \mathbb{R}$

1: **for** epoch $\in [1, ...n_e]$ **do**
2:   **for all** batch $\{(x_1, s_1, y_1), ..., (x_B, s_B, y_B)\}$ drawn from $\Gamma$ **do**
3:     **for** $i \in [1, ...n_g]$ **do**
4:       $J_s(\omega_f, \omega_g) \leftarrow \frac{1}{b} \sum_{i=1}^{b} l_S(g_{w_g}(f_{w_f}(x_i)), s_i)$ {Calculate the sensitive adversarial loss}
5:       $\omega_g \leftarrow \omega_g - \alpha_g \frac{\partial J_s(\omega_f, \omega_g)}{\partial \omega_g}$ {Update the sensitive adversarial model $g_{\omega_g}$ by gradient descent}
6:     **end for**
7:     **for** $j \in [1, ...n_r]$ **do**
8:       $\forall s \in \{0, 1\}$   $r_{w_r}(x_i, s_i = s) \leftarrow \frac{\sum_{i=1}^{b} e^{h_{w_r}(x_i, s_i)}}{\sum_{i=1}^{b} e^{h_{w_r}(x_i, s_i)} \mathbb{1}_{s_i = s}}$ {Normalize the likelihood ratio per demographic subgroups}
9:       $J_r(\omega_r, \omega_f, \omega_g) \leftarrow \frac{1}{b} \sum_{i=1}^{b} r_{w_r}(x_i, s_i) * l_S(g_{w_g}(f_{w_f}(x_i)), s_i) + \tau * r_{w_r}(x_i, s_i) \log(r_{w_r}(x_i, s_i))$ {Calculate the likelihood ratio cost function}
10:      $\omega_r \leftarrow \omega_r - \alpha_r \frac{\partial J_r(\omega_r, \omega_f, \omega_g)}{\partial \omega_r}$ {Update the adversarial model $r_{\omega_r}$ by gradient descent}
11:     **end for**
12:    $\forall s \in \{0, 1\}$   $r_{w_r}(x_i, s_i = s) \leftarrow \frac{\sum_{i=1}^{b} e^{h_{w_r}(x_i, s_i)}}{\sum_{i=1}^{b} e^{h_{w_r}(x_i, s_i)} \mathbb{1}_{s_i = s}}$ {Normalize the likelihood ratio per demographic subgroups}
13:    $J_f(\omega_r, \omega_f, \omega_g) \leftarrow \frac{1}{b} \sum_{i=1}^{b} l_Y(f_{w_f}(x_i), y_i) - \lambda r_{w_r}(x_i, s_i) * l_S(g_{w_g}(f_{w_f}(x_i)), s_i)$ {Calculate the loss function of the predictor model}
14:    $\omega_f \leftarrow \omega_f - \alpha_f \frac{\partial J_f(\omega_r, \omega_f, \omega_g)}{\partial \omega_f}$ {Update the predictor model}
15:   **end for**
16: **end for**

---

### A.9.2 BROAD ALGORITHM

The BROAD algorithm describes the non-parametric, implementation for the analytical solution of the problem considered in Eq 13. The main difference between the two algorithms is that BROAD does not include the update step for the ratio model $r_{w_r}$ and instead calculates the likelihood ratio directly in the predictor model update step. While it simplifies the optimization problem, it has been observed that the uncertainty sets generated by non-parametric approaches are pessimistic (i.e., may encompass the considered distributions), and as a result, lead to sub-optimal solutions.

This non-parametric algorithm, for the Demographic Parity objective, aims to train a locally fair model by iteratively updating two components: the sensitive adversarial model $g_{w_g}$ and the predictor model $f_{w_f}$. The required inputs for this algorithm are the same as those for the previous one.

- Update the sensitive adversarial model $g_{w_g}$: This step is the same as in ROAD. The sensitive adversarial loss is calculated based on the current parameters of the predictor and sensitive adversarial models. The sensitive adversarial model is then updated using traditional gradient descent.

- Calculate the likelihood ratio: Instead of updating the ratio model $r_{w_r}$ as the ROAD algorithm, the likelihood ratio is calculated directly using the exponential of the negative sensitive adversarial loss divided by the temperature parameter $\tau$. The likelihood ratio is normalized across sensitive groups.

- Update the predictor model: The loss function for the predictor model $f_{w_f}$ is calculated, taking into account the traditional loss function of the predictor and the average of the product between the weighting ratio and sensitive adversarial. The predictor model is then updated using gradient descent.

The process is repeated for the specified number of epochs. By iteratively updating these two models, the algorithm aims to achieve a balance between minimizing the loss function and maintaining fairness across demographic subgroups in the resulting predictor model.

In summary, the main difference between the two algorithms is the approach to updating the ratio model. ROAD updates the ratio model $r_{w_r}$ through an iterative process, while BROAD calculates the likelihood ratio directly in the predictor model update step. This alternative algorithm simplifies the process and may result in faster computation times. However, when comparing the performance with respect to fairness trade-offs, it has been observed to underperform compared to the original algorithm (ROAD).

---

**Algorithm 2** BROAD: Boltzmann Robust Optimization for Adversarial Debiasing (Conditional normalization)

---

**Require:** Training set $\Gamma$,

  Individual loss functions $l_Y$ and $l_S$,

  Batchsize $b$, Number of epochs $n_e$, Neural Networks $f_{\omega_f}$ and $g_{\omega_g}$,

  Learning rates $\alpha_f$, $\alpha_g$ and $\alpha_r$, Number of training iterations $n_g$,

  Fairness control $\lambda_g \in \mathbb{R}$ and Temperature control $\tau \in \mathbb{R}$

1: **for** epoch $\in [1, ...n_e]$ **do**

2:     **for all** batch $\{(x_1, s_1, y_1), ..., (x_B, s_B, y_B)\}$ drawn from $\Gamma$ **do**

3:         **for** $i \in [1, ...n_g]$ **do**

4:             $J_s(\omega_f, \omega_g) \leftarrow \frac{1}{b} \sum_{i=1}^{b} l_S(g_{w_g}(f_{w_f}(x_i)), s_i)$ {Calculate the sensitive adversarial loss}

5:             $\omega_g \leftarrow \omega_g - \alpha_g \frac{\partial J_s(\omega_f, \omega_g)}{\partial \omega_g}$ {Update the sensitive adversarial model $g_{\omega_g}$ by gradient descent}

6:         **end for**

7:         $\forall s \in \{0, 1\} \quad r(x_i, s_i) = \frac{e^{-\mathcal{L}_S(g_{w_g}(f_{w_f}(x_i)), s_i)/\tau}}{\frac{1}{n_{s=s_i}} \sum_{(x_j, s_j) \in \Gamma, s_j = s_i}^{n} e^{-\mathcal{L}_S(g_{w_g}(f_{w_f}(x_j)), s_j)/\tau}}$ {Calculate the likelihood ratio per demographic subgroups}

8:         $J_f(\omega_f, \omega_g) \leftarrow \frac{1}{b} \sum_{i=1}^{b} l_Y(f_{w_f}(x_i), y_i) - \lambda r(x_i, s_i) * l_S(g_{w_g}(f_{w_f}(x_i)), s_i)$ {Calculate the loss function of the predictor model}

9:         $\omega_f \leftarrow \omega_f - \alpha_f \frac{\partial J_f(\omega_f, \omega_g)}{\partial \omega_f}$ {Update the predictor model}

10:     **end for**

11: **end for**

---

## A.10 ADAPTING ROAD TO EQUALIZED ODDS

As presented in the section2 Equalized Odds (EO) (Hardt et al., 2016) is another group fairness criterion that requires the classifier to have equal true positive rates (TPR) and false positive rates (FPR) across demographic groups. This criterion is especially relevant when misclassification can have significant impacts on individuals from different groups. The constraint of equalized odds can be expressed as follows, where $\epsilon_1$ and $\epsilon_2$ represent the allowed deviation from perfect TPR denoted as $\Delta_{TPR}$ and FPR $\Delta_{FPR}$ denoted as respectively.

$$\min_{w_f} \quad \mathbb{E}_{(x,y,s)\sim p} \, \mathcal{L}_\mathcal{Y}(f_{w_f}(x), y)$$

$$\text{s.t.} \quad |\mathbb{E}_{(x,y,s)\sim p}(\hat{f}_{w_f}(x)|y=1, s=1) - \mathbb{E}_{(x,y,s)\sim p}(\hat{f}_{w_f}(x)|y=1, s=0)| < \epsilon_1 \quad (10)$$

$$|\mathbb{E}_{(x,y,s)\sim p}(\hat{f}_{w_f}(x)|y=0, s=1) - \mathbb{E}_{(x,y,s)\sim p}(\hat{f}_{w_f}(x)|y=0, s=0)| < \epsilon_2$$

As explained in section 2.1, to achieve EO Zhang et al. (2018) employs an adversarial learning approach by concatenating the true outcome $Y$ to the input of the adversary. The resulting relaxed optimization problem can be expressed as $\min_{w_f} \max_{w_g} \mathbb{E}_p \, \mathcal{L}_\mathcal{Y}(f(x), y) - \lambda \mathbb{E}_p \mathcal{L}_\mathcal{S}(g_{w_g}(f_{w_f}(x), y), s)$. This methodology balances minimizing the predictor model's loss and maximizing the adversarial model's loss while satisfying equalized odds.

As for demographic parity, we propose to optimize accuracy while adhering to a worst-case equalized odds constraint. We define the subpopulations of interest, for which we aim to optimize fairness, as distributions $q$ within an *uncertainty set* $\mathcal{Q}$, and present the DRO framework for the Equalized Odds criterion as follows:

$$\min_{w_f} \quad \mathbb{E}_{(x,y,s)\sim p} \mathcal{L}_\mathcal{Y}(f_{w_f}(x), y)$$

$$\text{s.t.} \quad \max_{q \in \mathcal{Q}} \left| \mathbb{E}_{(x,y,s)\sim q} \left[ \hat{f}_{w_f}(x)|y=1, s=1 \right] - \mathbb{E}_{(x,y,s)\sim q} \left[ \hat{f}_{w_f}(x)|y=1, s=0 \right] \right| < \epsilon_1, \quad (11)$$

$$\left| \mathbb{E}_{(x,y,s)\sim q} \left[ \hat{f}_{w_f}(x)|y=0, s=1 \right] - \mathbb{E}_{(x,y,s)\sim q} \left[ \hat{f}_{w_f}(x)|y=0, s=0 \right] \right| < \epsilon_2$$

The constraint ensures that the False Positive Rate and the False Negative Rate between the two sensitive groups remain less than a predefined threshold $\epsilon_1$ and $\epsilon_2$ respectively under the worst-case distribution $q \in \mathcal{Q}$. Working with distribution $q$ allows us to enforce local fairness by targeting

subpopulations of interest, thus creating a more focused and adaptable model that addresses fairness problems both globally and at a granular level. As for the demographic parity task we restrict $\mathcal{Q}$ to the set of distributions that are absolutely continuous with respect to $p$. This allows us to write $q = rp$, with $r : \mathcal{X} \times \mathcal{S} \times \mathcal{Y} \to \mathbb{R}^+$ a function that acts as a weighting factor.

Given a training set $\Gamma$ sampled from $p$, we can thus reformulate the overall objective, by substituting $q$ with $rp$ and applying its Lagrangian relaxation, as an optimization problem on $r \in \mathcal{R} = \{r \mid rp \in \mathcal{Q}\}$:

$$\min_{w_f} \max_{r \in \mathcal{R}} \frac{1}{n} \sum_{i=1}^{n} \mathcal{L}_{\mathcal{Y}}(f_{w_f}(x_i), y_i) - \lambda_g \frac{1}{n} \sum_{i=1}^{n} r(x_i, s_i, y_i) \mathcal{L}_{\mathcal{S}}(g_{w_g^*}(f_{w_f}(x_i), y_i), s_i) \quad (12)$$

$$\text{with } w_g^* = \arg\min_{w_g} \frac{1}{n} \sum_{i=1}^{n} \mathcal{L}_{\mathcal{S}}(g_{w_g}(f_{w_f}(x_i), y_i), s_i)$$

With $\lambda_g$ a regularization parameter controlling the trade-off between accuracy and fairness in the predictor model.

The overall optimization problem of our Robust Optimization for Adversarial Debiasing (ROAD) framework can thus finally be formulated as (full derivation given in A.1):

$$\min_{w_f} \max_{r \in \bar{\mathcal{R}}} \frac{1}{n} \sum_{i=1}^{n} \mathcal{L}_Y(f_{w_f}(x_i), y_i) - \lambda_g \Big[ \frac{1}{n} \sum_{i=1}^{n} r(x_i, s_i, y_i) \mathcal{L}_{\mathcal{S}}(g_{w_g^*}(f_{w_f}(x_i), y_i), s_i)$$

$$+ \tau \underbrace{\frac{1}{n} \sum_{i=1}^{n} r(x_i, s_i, y_i) \log(r(x_i, s_i, y_i))}_{\text{KL constraint}} \Big] \quad (13)$$

$$\text{with } w_g^* = \arg\min_{w_g} \frac{1}{n} \sum_{i=1}^{n} \mathcal{L}_S(g_{w_g}(f_{w_f}(x_i), y_i), s_i)$$

As for the demographic parity task, to introduce more local smoothness in the fairness weights assigned to training samples, we propose an implementation of the $r$ function via a neural network architecture. Our goal is to ensure that groups of similar individuals, who might be neglected in the context of group fairness mitigation (e.g., due to their under-representation in the training population, cf. Fig. 1), receive a similar level of attention during the training process.

To enforce the conditional validity constraint presented earlier, we employ an exponential parametrization with two batch-level normalizations, one for each demographic group. For each sample $(x_i, y_i, s_i)$ in the mini-batch, we define the normalized ratio as:

$$\forall i, r_{w_r}(x_i, s_i, y_i) = \frac{e^{h_{w_r}(x_i, s_i, y_i)}}{\frac{1}{n_{s_i}} \sum_{(x_j, s_j, y_j) \in \Gamma, s_j = s_i} e^{h_{w_r}(x_j, s_j, y_j)}}$$

with $h : \mathcal{X} \times \{0; 1\} \to \mathcal{R}$ a neural network with weights $w_r$.

To train ROAD, we use an iterative optimization process, alternating between updating the predictor model's parameters $w_f$ and updating the adversarial models' parameters $w_g$ and $w_r$ by multiple steps of gradient descent.

To obtain better results, as commonly done in the literature, we perform several training iterations of the adversarial networks for each prediction iteration. This approach results in more robust adversarial algorithms, preventing the predictor classifier from dominating the adversaries.

The required inputs for ROAD include the training data containing $X$, $Y$, and $S$ (please note that $S$ is not required at the testing time), the loss functions (i.e., logloss function applied in our experiment), batch size, number of epochs, neural networks architectures, learning rates, and the control parameters ($\tau$ and $\lambda$).

For each epoch, the algorithm 1 iterates through all the batches in the training data. In each batch, the algorithm performs the following steps:

- Update the sensitive adversarial model $g_{w_g}$: The sensitive adversarial takes the current output predictions $f_{w_f}(x)$ as input. The sensitive adversarial loss is calculated based on the current parameters of the predictor and sensitive adversarial models. The sensitive adversarial model is then updated using traditional gradient descent.

- Update the ratio model $r_{w_r}$: The likelihood ratio is normalized across demographic subgroups. The likelihood ratio cost function is calculated with respect to the average of the product between the weighting ratio and sensitive adversarial loss in addition to the KL constraint. Then, the adversarial model is updated using gradient descent.

- Update the predictor model: The loss function for the predictor model $f_{w_f}$ is calculated, taking into account the traditional loss function of the predictor and the average of the product between the weighting ratio and sensitive adversarial. The predictor model is then updated using gradient descent.

The process is repeated for the specified number of epochs. By iteratively updating these three models, the algorithm aims to achieve a balance between minimizing the loss function and maintaining fairness across demographic subgroups in the resulting predictor model.

---

**Algorithm 3** ROAD for Equalized Odds

---

**Require:** Training set $\Gamma$,
    Individual loss functions $l_Y$ and $l_S$,
    Batchsize $b$, Number of epochs $n_e$, Neural Networks $r_{\omega_r}$, $f_{\omega_f}$ and $g_{\omega_g}$,
    Learning rates $\alpha_f$, $\alpha_g$ and $\alpha_r$, Number of training iterations $n_g$ and $n_r$,
    Fairness control $\lambda_g \in \mathbb{R}$ and Temperature control $\tau \in \mathbb{R}$

1: **for** epoch $\in [1,...n_e]$ **do**
2:   **for all** batch $\{(x_1,s_1,y_1),...,(x_B,s_B,y_B)\}$ drawn from $\Gamma$ **do**
3:     **for** $i \in [1,...n_g]$ **do**
4:       $J_s(\omega_f,\omega_g) \leftarrow \frac{1}{b}\sum_{i=1}^{b} l_S(g_{w_g}(f_{w_f}(x_i),y_i),s_i)$ {Calculate the sensitive adversarial loss}
5:       $\omega_g \leftarrow \omega_g - \alpha_g \frac{\partial J_s(\omega_f,\omega_g)}{\partial \omega_g}$ {Update the sensitive adversarial model $g_{\omega_g}$ by gradient descent}
6:     **end for**
7:     **for** $j \in [1,...n_r]$ **do**
8:       $\forall s \in \{0,1\}$   $r_{w_r}(x_i,s_i=s,y_i) \leftarrow \frac{\sum_{i=1}^{b} e^{h_{w_r}(x_i,s_i,y_i)}}{\sum_{i=1}^{b} e^{h_{w_r}(x_i,s_i,y_i)}\mathbb{1}_{s_i=s}}$ {Normalize the likelihood ratio per demographic subgroups}
9:       $J_r(\omega_r,\omega_f,\omega_g) \leftarrow \frac{1}{b}\sum_{i=1}^{b} r_{w_r}(x_i,s_i,y_i) * l_S(g_{w_g}(f_{w_f}(x_i),y_i),s_i) + \tau * r_{w_r}(x_i,s_i,y_i)\log(r_{w_r}(x_i,s_i,y_i))$ {Calculate the likelihood ratio cost function}
10:      $\omega_r \leftarrow \omega_r - \alpha_r \frac{\partial J_r(\omega_r,\omega_f,\omega_g)}{\partial \omega_r}$ {Update the adversarial model $r_{\omega_r}$ by gradient descent}
11:     **end for**
12:    $\forall s \in \{0,1\}$   $r_{w_r}(x_i,s_i=s,y_i) \leftarrow \frac{\sum_{i=1}^{b} e^{h_{w_r}(x_i,s_i,y_i)}}{\sum_{i=1}^{b} e^{h_{w_r}(x_i,s_i,y_i)}\mathbb{1}_{s_i=s}}$ {Normalize the likelihood ratio per demographic subgroups}
13:    $J_f(\omega_r,\omega_f,\omega_g) \leftarrow \frac{1}{b}\sum_{i=1}^{b} l_Y(f_{w_f}(x_i),y_i) - \lambda r_{w_r}(x_i,s_i,y_i) * l_S(g_{w_g}(f_{w_f}(x_i),y_i),s_i)$ {Calculate the loss function of the predictor model}
14:    $\omega_f \leftarrow \omega_f - \alpha_f \frac{\partial J_f(\omega_r,\omega_f,\omega_g)}{\partial \omega_f}$ {Update the predictor model}
15:   **end for**
16: **end for**

---

## A.11 LIMITATIONS

In this section, we discuss a few limitations of our work.

### A.11.1 TRADITIONAL GROUP FAIRNESS LIMITATIONS.

Several well-known limitations may hinder the efficacy of group fairness methods. In that regard, our approach may suffer equally.

**Unavailability of the sensitive attribute.** The majority of group fairness methods rely on the sensitive attribute $S$ being available. This is also the case for our method, as $S$ is used as a target variable for the adversary $g$. Although a few works have proposed methods to circumvent the issue (Hashimoto et al., 2018; Lahoti et al., 2020), this remains a major problem for algorithmic fairness, as many real-world situations fall into this category (Tomasev et al., 2021)

**Impossibility of Fairness** The "impossibility of fairness" refers to trade-offs that exist between fairness criteria (Chouldechova, 2017; Kleinberg et al., 2017). In practice, this means that criteria such as Demographic Parity and Equalized Odds cannot be fully satisfied at the same time. More generally, a strong dependence between the sensitive attribute $S$ and the target variable $Y$ has been shown to make the task enforcing fairness without sacrificing too much accuracy difficult.Like other group fairness methods, our proposed approach suffers from the same limitations.

### A.11.2 LIMITATIONS TO THE NOTION OF LOCAL FAIRNESS

One of the contributions of our work is the introduction of the notion of local fairness. Although intuitive, local fairness is difficult to formalize exactly, as partly discussed in Section 2.2. We see two main obstacles hurting the adoption of the notion in practice:

- Although subgroups of interest do not need to be defined at training time, they obviously need to be when evaluating the approach. For this purpose, experts need to be able to define, and agree, on an exact definition for these subgroupes. Yet, as discussed by Wachter et al. (2021), this is sometimes difficult in practice. In their work, Wachter et al. thus give examples of several European regulatory bodies (e.g. European Court of Justice and European Parlement) disagreeing on definitions of gender, age groups, or work categories.
- Group fairness criteria, such as Demographic Parity and Equalized Odds, are measured in all subgroups. For these measures to be significant, the subgroups need to be of a certain minimum size. Yet, this minimum size may be difficult to determine.

Although ROAD was proposed as a solution circumventing these issues, we still believe these issues may hurt a potential widespread adoption of the notion.

### A.11.3 LIMITATIONS TO THE PROPOSED METHOD (ROAD)

Besides the sensitive attribute being unavailable, another limitation for our approach could be when the subpopulations of interest in specific usecases would be described with features that are unavailable. For instance, when enforcing fairness in the context of text (e.g. tweet) toxicity prediction, one might be interested in making sure that the model is fair (e.g. w.r.t. to employed racial dialect Ball-Burack et al. (2021)) not only globally, but also that this fairness holds independently of the, say, the context of the tweet. Yet, this context might not always be available.

