# OpenReview forum: "On the Fairness ROAD: Robust Optimization for Adversarial Debiasing"
_ICLR.cc/2024/Conference — ICLR 2024 poster_

### Official Review · Reviewer_Cpzk · 2023-10-30

**Soundness:** 3 good
**Presentation:** 3 good
**Contribution:** 2 fair
**Rating:** 6
**Confidence:** 5

**Summary:**

This paper addresses the problem of local fairness, which ensures the model fairness on the subregion split via unknown attributes.  They proposed an approach based on the idea of distributional robust optimization, under an adversarial learning paradigm. The desired fairness is implemented of instance-wise sample reweights and the experiments are conducted to demonstrate the proposed method can achieve Pareto dominance.

**Strengths:**

The local fairness introduced in this paper is interesting, and the analysis to existing literature is almost thorough and clear. The method looks good, and the writing is well prepared.

**Weaknesses:**

I have some doubts about concepts and formulations after reading the paper. Also, the technical contribution may need further clarifications.

**Questions:**

1.	The motivation of this paper is that global fairness methods usually cannot guarantee a consistent fairness value on split subregions, shown as Fig. 1. However, such observations are based on fine-grained partition (e.g., age categories) of training data and each subregion might be with a relatively small size. In this case, the disparity tends to happen. For example, DI might be very large if a bin only includes two samples. How do we confirm that biases do widely exist and then precisely define “local fairness”?
2.	Following question 1, I think it might be necessary to use some prior for split subregions. For example, in DRO, they introduced a lower bound of the subgroup ratio. With accuracy as a utility, BPF demonstrated that very small group would lead to uniform classifier. But from sec 3.2, as Softmax weights can be regarded as applying a Shannon entropy regularization. It would be better if the authors can point out such behind insights.

        Blind Pareto Fairness and Subgroup Robustness, ICML 2021.

3.	Starting from Eq. (2), the input of $g_{w_g}(.)$ directly takes the output of $f_{w_f}(.)$. So, how do you implement $w_g$? Considering a binary case, $f_{w_f}(.)$ is a only probability in output space.
4.	Following question 3, please check the constraint of eq. (4), where s should be properly positioned.
5.	Above Eq. (4), why distribution $q$ can help characterize local fairness is not clear for me. Please note that in DRO, it serves any subregion’s upper bound.
6.	Since $\lambda_g$ is not learnable (Sec A.5.4), are the best results selected to have a Pareto dominance as claimed?
7.	Please justify the novelty/contribution compared to Michel et al 2022.

---

> ### Author Response · Authors · 2023-11-17
>
> > 1. The motivation of this paper is that global fairness methods usually cannot guarantee a consistent fairness value on split subregions, shown as Fig. 1. However, such observations are based on fine-grained partition (e.g., age categories) of training data and each subregion might be with a relatively small size. In this case, the disparity tends to happen. For example, DI might be very large if a bin only includes two samples. How do we confirm that biases do widely exist and then precisely define “local fairness”?
>
>
>
> While our definition of local fairness is given in Eq.(3) as means of expectations for a specific family of $q$ distributions, we acknowledge that measuring fairness from finite datasets may face difficulties. Given a considered partition of $X$, the presence of small subgroups can easily result in large DI values, and hence, it is essential to define a threshold to properly demonstrate the existence of the local fairness problem. This threshold is to be defined depending on the use case, and represents the minimum group size for a measure of local fairness to be deemed significant.
>
> In our work, we do use such thresholds when measuring local fairness: we highlight the existence of the problem (Figure 1) by considering a minimum of 50 individuals in each represented age category (Adult and COMPAS). This threshold allowed us to observe significant disparity between sensitive groups, and conclude on the existence of the local unfairness problem. The local fairness results presented in Figure 2 and Figure 5 follow the same methodology: subgroups of minimum 50 individuals for the COMPAS dataset, and 20 for the Law and German datasets (details for all subgroups are given in Appendix A.5.3).
>
> Besides, to make sure that the problem of local fairness we observe is not an artifact of overfitting a threshold, we have conducted multiple experiments to test the robustness of our observations: in Figure 5, Figure 9, and Figure 10, we test the behavior of models trained once when this threshold varies. These experiments show that no matter the value of the threshold chosen (among a reasonable range), local bias problems could be observed.
>
> Finally, we would like to restate that these thresholds are only needed at evaluation time, to measure local fairness. During the training, no threshold is needed, as the locality of the attention is controlled by the temperature $\tau$ hyperparameter.
>
>
>
> > 2 .Following question 1, I think it might be necessary to use some prior for split subregions. For example, in DRO, they introduced a lower bound of the subgroup ratio. With accuracy as a utility, BPF demonstrated that very small group would lead to uniform classifier. But from sec 3.2, as Softmax weights can be regarded as applying a Shannon entropy regularization. It would be better if the authors can point out such behind insights.
>   Blind Pareto Fairness and Subgroup Robustness, ICML 2021.
>
> Thanks to the reviewer for this very insightful remark. Since considering worst-case distributions, mentionned works [1] and [2] have indeed connections with our work, that we could discuss.
>
> The DRO approach [1], as [3], assumes P as an unknown mixture of distributions and attempt to minimize the risk of the worst components of that mixture. Using the dual form of the problem, they consider an upperbound of the risk that depends on the marginal of the smallest component of the mixture. To avoid degenerating through trivial solutions that outputs uniform predictions, they indeed set a minimal threshold for the size of that smallest component they could focus on. This threshold is shown to be related to the maximal divergence allowed between their worst-case distribution $q$ and the training one $p$.
>
> Please note that their setting is different than ours: their goal is to train models that perform uniformly well across all partitions of the population, while ours is to train a model that is uniformly fair (regarding a sentitive attribute) accross all subregions of the feature space, which is quite different. In particular, they usually consider convex losses and contraints to build their models, while we envison more complex architectures designed through neural networks. However, setting a prior on the  least populated group for which we can ensure fairness can be seen equivalent as tuning our hyper-parameter $\tau$, that controls the Shannon entropy of our divergence ratio $r$ (defined as a maximum entropy distribution using softmax weights).
>
> We thank again the reviewer for their comments, that led us to include several components of the above discussion in the paper in sections 2.2, 2.3 and 3.
>
>
> [1] John Duchi, Tatsunori Hashimoto, and Hongseok Namkoong. Distributionally robust losses for latent covariate mixtures. Operations Research, 71(2):649–664, 2023.

---

> > ### Author Response · Authors · 2023-11-17
> >
> > [2] Natalia L Martinez, Martin A Bertran, Afroditi Papadaki, Miguel Rodrigues, and Guillermo Sapiro. Blind pareto fairness and subgroup robustness. In International Conference on Machine Learning, pp. 7492–7501. PMLR, 2021.
> >
> > [3] Tatsunori Hashimoto, Megha Srivastava, Hongseok Namkoong, and Percy Liang. Fairness without demographics in repeated loss minimization. In International Conference on Machine Learning, pp. 1929–1938. PMLR, 2018.
> >
> > > 3. Starting from Eq. (2) the input of g_{w_g}(.) directly takes the output of f_{w_f}(.). So, how do you implement w_g? Considering a binary case, f_{w_f}(.) is a only probability in output space.
> >
> > As noted by the reviewer, the sensitive adversarial $g_{w_g}(.)$  takes only one variable as input, which is the current output predictions $f_{w_f}(x)$. To reconfirm, it is important to note that our approach utilizes the (continuous) probability output $f_{w_f}(.)$, NOT the predicted label (after threshold) $\hat{f}$. This is a common practice in adversarial fairness reconstruction, as demonstrated in works such as [Zhang et al., 2018, Adel et al. 2019], etc. and helps ensuring differentiability of the objective.  As discussed in the second paragraph of appendix A.2.2, the adversary is then considered optimal for Demographic Parity when it accurately outputs $\mathbb{E}[S|f(X)]$ for any input. A classifier is deemed fully fair when $\mathbb{E}[S|f(X)] = P(S=1|f(X)) = P(S=1)$. In this case, we have $P(S=1, f(X)) = P(S=1)P(f(X))$, which indicates independence and, consequently, $P(\hat{f}(X)|S) = P(\hat{f}(X))$.
> >
> > Please note that while this is only one input, state-of-the-art (SOTA) methods have shown the necessity to segment this input in several ways to better capture information relevant to predicting the sensitive attribute[1].
> >
> > Furthermore to obtain better results, as commonly done in the literature, we perform several training iterations of the adversarial networks for each prediction iteration. This approach results in more robust adversarial algorithms, preventing the predictor classifier from dominating the adversary (more details can be found in appendix in section A.9.).
> >
> > [1] Fair adversarial gradient tree boosting. 2019 IEEE International Conference on Data Mining (ICDM). IEEE, 2019.
> >
> >
> > > 4. Following question 3, please check the constraint of eq. (4), where s should be properly positioned.
> >
> > Thank you for your comment. We acknowledge that there was indeed an issue in Eq. (4), as a bracket was missing. To clarify, we have modified the equation from $\mathcal{L_S}(g_{w_g}(f_{w_f}(x), s))$ to $\mathcal{L_S}(g_{w_g}(f_{w_f}(x)), s)$, ensuring that s is correctly positioned, as it is in the traditional formulation by Zhang et al., 2018. We have updated the paper accordingly.
> >
> >
> > > 5. Above Eq. (4), why distribution q can help characterize local fairness is not clear for me. Please note that in DRO, it serves any subregion’s upper bound.
> > >
> >  Please note that DRO for accuracy bounds the worst-case accuracy over groups while DRO for fairness constrains the worst-case local fairness. Thus, we are solving for an upper bound on local unfairness as long as $\cal{Q}$ contains subpopulations of interest. Please also note that the Lipschitzness of the neural networks $r$ introduces implicit locality smoothness assumption in the input space, which helps define the distributions q as subregions within the feature space. Discrepancies of q regarding p cannot be spread on disconnected individuals, but are driven by locality in the feature space.
> >
> >
> > > 6. Since λ_g is not learnable (Sec A.5.4), are the best results selected to have a Pareto dominance as claimed?
> >
> > In response to the concern regarding the selection of the best results for Pareto dominance in the presence of a non-learnable λ_g parameter, we would like to emphasize the following points:
> >
> >
> > 1) It is common in algorithmic fairness literature to have a non-learnable λ_g parameter, which allows for the tradeoff between fairness and accuracy. The Pareto dominance is frequently used to demonstrate this trade-off, as shown in Wang et al. TMLR 2023 [1], Ferry et al. 2021 [2], and Chai et al. [3]. This approach enables the examination of a wide range of fairness and accuracy levels.
> > 2) In our experiments, we examined an extensive range of λ values, including large ones (e.g., λ=5), which generally represent the upper limit for the algorithm without encountering empirical degeneration. Reported results correspond to non-dominated solutions w.r.t. both criteria.
> >
> > [1] Wang, Haotao, et al. “How Robust is Your Fairness? Evaluating and Sustaining Fairness under Unseen Distribution Shifts.” Transactions on machine learning research 2023 (2023).
> >
> > [2] Ferry, Julien, et al. “Improving fairness generalization through a sample-robust optimization method.” Machine Learning 112.6 (2023): 2131-2192.
> >
> > [3] Chai, Junyi, and Xiaoqian Wang. “Fairness with adaptive weights.” International Conference on Machine Learning. PMLR, 2022.

---

> > > ### Author Response · Authors · 2023-11-17
> > >
> > > > 7. Please justify the novelty/contribution compared to Michel et al 2022.
> > >
> > > In response to the reviewer’s question regarding the novelty and contributions of our work compared to [Michel et al. 2022], we would like to highlight the following key differences:
> > >
> > > **Different problem formulation**: Our approach diverges significantly from [Michel et al. 2022] in terms of the problem addressed. While [Michel et al. 2022] focus on parametric DRO applied to the training objective $L_Y$ without considering any fairness criterion (i.e., no sensitive attribute $S$ is used), our work aims to optimize fairness criteria, such as Demographic Parity. This distinction in objectives is crucial, as optimizing fairness criteria is not achievable through the methods proposed in [Michel et al. 2022]. This distinction is now emphasized in section 2.2, with DRO methods for fairness without demographics, following your suggestions in question 2 above.
> > >
> > > **Different optimization**: The traditional DRO pursued by [Michel et al. 2022] aims to minimize the maximum training loss, which contrasts with our opposite goal of maximizing the minimum possible adversarial loss. In our approach, we assign more weight to samples that are easier for the adversary to reconstruct $S$  instead of focusing on less classifier accuracy as in [Michel et al. 2022]. Additionally, our optimization involves three neural networks, as opposed to the two in [Michel et al. 2022], and the validity constraint differs between the two works. We want to emphasize that our technique employs a multi-objective optimization approach that balances fairness and accuracy, while [Michel et al. 2022] focus primarily on optimizing accuracy.
> > >
> > > Furthermore,  to address your concerns, we have compared our method to [Michel et al. 2022]. In Figure 11 (Appendix A.5: “Additional results on the difference with traditional DRO,”), is shown the Pareto front for the experiment on Local Fairness for the German dataset. Our approach consistently achieves better results in the Pareto front, demonstrating improved performance in terms of both fairness and accuracy compared to existing methods.

---

> > > > ### Comment · Reviewer_Cpzk · 2023-11-23
> > > > **Thanks for the responses**
> > > >
> > > > Thanks for responding my questions. Your answers are clear and I would like to raise my rate score to 6.  For your explanation "In our approach, we assign more weight to samples that are easier for the adversary to reconstruct $S$ instead of focusing on less classifier accuracy as in [Michel et al. 2022]. " I agreed with this point. However, from my view, such reweighing strategy in an adversarial manner for fairness research is quit common according to literature, and I personally would see more exciting methods that could deal with such problems in top AI conferences.

---

### Official Review · Reviewer_34eA · 2023-10-30

**Soundness:** 3 good
**Presentation:** 3 good
**Contribution:** 3 good
**Rating:** 8
**Confidence:** 4

**Summary:**

This paper focuses on a notion of local fairness as opposed to group fairness. When local subgroups are unknown at training time, it is difficult to enforce local fairness constraints. This paper expands the method DRO to include an adversarial learning component that specifically minimizes the ability of an adversary to reconstruct sensitive attributes. The main idea is to boost the importance of regions where sensitive reconstruction is easiest. This is done at each optimization step. To do this efficiently, the paper introduces an adversarial importance re-weighting method. They provide the analytical formulation for this method as well as two implementations (one non-parametric and one parametric). Finally, they experimentally evaluate the proposed method including ablation studies.

**Strengths:**

(1) The main contribution is novel and clearly specified. Combining DRO with adversarial learning is simple yet effective.

(2) The paper is written with clarity and exposition is easy to follow throughout. Problem statement as well as the two implementations are concise.

**Weaknesses:**

(1) The paper does not directly address limitations. For instance, is there a computational scalability issue with this technique compared to other more well known group fairness methods?

(2) The reference to applied fairness research could be strengthened. There are many related works that are omitted. In particular, work on fairness and adversarial learning.

**Questions:**

(1) Is there a reason why the first paragraph of the introduction omits citations in statements like "models...have been shown to unintentionally perpetuate existing biases and discriminations"?

(2) What do you see as the main use case for this method? What are the limitations you anticipate other than sensitive attribute not being available?

---

> ### Author Response · Authors · 2023-11-17
>
> > (1) The paper does not directly address limitations. For instance, is there a computational scalability issue with this technique compared to other more well known group fairness methods?
>
> Thank you for your comment. In terms of computational scalability, the training process of our method is relatively stable, and the networks do not require many neurons (3 layers with no more than 64 neurons per layer). Our study in the appendix A.4.1, titled “Does the complexity of the two adversarial networks have an impact on the quality of the results?” shows that for network r, a simpler linear prediction such as logistic regression is quite efficient (please note that due to the exponential parameterization, an exponential activation is applied to it). We believe that our method strikes a balance between complexity and the potential improvements in local fairness it offers. For limitations, see our answer to Question 2 below.
>
> > (2) The reference to applied fairness research could be strengthened. There are many related works that are omitted. In particular, work on fairness and adversarial learning.
>
> We appreciate the suggestion to strengthen the references to applied fairness research and related works. We have included the following references in the manuscript:
> Applied fairness:
>
> [5] Dissecting racial bias in an algorithm used to manage the health of populations, Obermeyer 2019
>
> [6] Achieving Fairness through Adversarial Learning: An Application to Recidivism Prediction, Wadsworth et al. 2018
>
> [7] A Fair Pricing Model via Adversarial Learning, Grari et al. 2022
>
>
> Some missing related works on adversarial learning for fairness:
>
> [8] Achieving Fairness through Adversarial Learning: An Application to Recidivism Prediction, Wadsworth et al. 2018
>
> [9] Learning to Pivot with Adversarial Networks, Louppe et al. 2017
>
>
> > (1) Is there a reason why the first paragraph of the introduction omits citations in statements like "models...have been shown to unintentionally perpetuate existing biases and discriminations"?
>
>
> Thank you for your comment. In our revised manuscript, we have included more relevant literature to better situate our work within the broader context of fairness research. These include for instance:
>
> [10] Machine Bias, Angwin et al. 2016 (Propublica Compas study)
>
> [11] Weapons of Math Destruction, Cathy O’Neil, 2016
>
>
> > (2) What do you see as the main use case for this method? What are the limitations you anticipate other than sensitive attribute not being available?
>
>
> *Usecases*
>
> Our contribution generally aims towards a better generalization of fairness for classification tasks. In this context, we believe it should be considered as an alternative in all problems and usecases already tackled by the field of algorithmic fairness: healthcare, credit obtention, etc.
> More particularly, several works in applied fairness have observed and discussed problems of “fairness generalization” or “local unfairness”. These usecases would thus represent an especially interesting area to explore. Some of them include:
>
> [12] focus on mitigating bias in a task of harmful tweet detection using several datasets. The sensitive feature is whether the tweet content is aligned with racial dialect (e.g. African American English). However, they observe that despite generally managing to enforce bias, the models showed poor generalization capabilities across datasets. In such situation, ROAD might help ensuring better fairness generalization.
>
> [13] observed that, in a context of car insurance pricing, disparities among groups persist in some geographical regions even after mitigating bias on the policyholder’s residence using adversarial methods.  Our approach can be a powerful tool in addressing such local fairness issues.
>
> [12] Differential Treatment: Mitigating Racial Dialect Bias in Harmful Tweet Detection, Ball-Burack et al. 2021
>
> [13] A Fair Pricing Model via Adversarial Learning [Grari et al. 2022]

---

> > ### Author Response · Authors · 2023-11-17
> >
> > *Limitations*
> >
> > Thank you for the interesting suggestion to add a discussion about limitations for our work. We have added a discussion on the topic in Section A.11. Below, we discuss some of them:
> >
> > Besides the sensitive attribute not being available, which the reviewer suggests, another limitation for our approach could be when the subpopulations of interest in specific usecases would be described with features that are unavailable. For instance, when enforcing fairness in the context of text (e.g. tweet) toxicity prediction, one might be interested in making sure that the model is fair (e.g. w.r.t. to employed racial dialect [14]) not only globally, but also that this fairness holds independently, for instance, of the context of the tweet. Yet, this context might not always be available.
> >
> > Besides, one of the contributions of our work is the introduction of the notion of local fairness. Although intuitive, local fairness is difficult to formalize exactly, as partly discussed in Section 2.3. We see two main obstacles hurting the adoption of the notion in practice:
> > 1) Although subgroups of interest do not need to be defined at training time, they obviously need to be when evaluating the approach. For this purpose, experts need to be able to define, and agree, on an exact definition for these subgroups. Yet, as discussed by Wachter et al. 2021, this is sometimes difficult in practice. In their work, Wachter et al. thus give examples of several European regulatory bodies (e.g. European Court of Justice and European Parlement) disagreeing on definitions for gender, age groups, or work categories.
> > 2) Group fairness criteria, such as Demographic Parity and Equalized Odds, are measured in all subgroups. For these measures to be significant, the subgroups need to be of a certain minimum size. Yet, this minimum size may be difficult to determine.
> >
> > Although ROAD was proposed as a solution circumventing these issues, we still believe these issues may hurt a potential widespread adoption of the notion of local fairness.
> >
> >
> > [14] Differential Treatment: Mitigating Racial Dialect Bias in Harmful Tweet Detection, Ball-Burack et al. 2021

---

> > > ### Comment · Reviewer_34eA · 2023-11-21
> > > **Thank you for the responses**
> > >
> > > Thank you for responding and answering all of my questions.

---

### Official Review · Reviewer_bLrb · 2023-11-02

**Soundness:** 3 good
**Presentation:** 4 excellent
**Contribution:** 3 good
**Rating:** 6
**Confidence:** 3

**Summary:**

Most prior work on group fairness measure the fairness discrepancy using global averages across pre-defined groups (e.g. males vs females).  However, in practice, one may care about fairness discrepancies in sub-regions of the feature space (e.g. males and females belonging to a particular age group). In this paper, the authors enforce fairness across such local sub-regions, by employing a form of distributionally robust optimization, equipped with an adversary that seeks to predict the sensitive attribute from the model predictions. Importantly, the proposed method does *not* assume the sub-regions to be known during training time, and instead optimizes over a set of possible sub-regions defined by an uncertainty set.

**Strengths:**

- A practically relevant problem statement with a clearly presented problem formulation
- The solution approach presented makes for a satisfactory contribution
- Strong experimental results
- Well-written paper

**Weaknesses:**

- The problem formulation seems *overly complex*, with multiple levels of nested minimization problems. It is unclear if the complexity is absolutely necessary (see question 1).
- Justifications needed for choice of uncertainty set $\mathcal{Q}$.
- *Lack of strong convergence guarantees* for the proposed optimization strategy. In fact, owing to the nested nature of the optimization problem, it appears (please correct me if I am incorrect) that it would be hard to provide theoretical guarantees even if under a simplistic scenario when $f$ and $g$ are linear models.

**Questions:**

(1) **Complexity of problem for formulation**: The authors seek to impose a robust version of the demographic parity constraint in (3), by introducing a maximization over an uncertainty set in the constraint. Instead of directly seeking to solve this problem, they formulate a equivalent problem with an additional argmin on an adversary's reconstruction loss (4). *Could authors elaborate why they chose to work with (4) instead of the simpler form in (3)?* It appears that the trick of applying instance-level weighting function $r$ could be applied as well to solving (3).

(2) **Assumptions on $q$**: Could the authors formally state the assumptions on the class of perturbed distributions $q$, preferably early on in the paper (e.g. Sec 2). My understanding is that the group priors are required to be the same for both q and p, i.e. $q(s) = p(s)$. However, its unclear if the validity constraint described in Sec 3.1 would include **all distributions $q$ for with the same group priors as $p$**. It might be cleaner to have the assumptions on $q$ described first, and then argue why the particular choice of weighting functions $r$ satisfy those assumptions.

(3) **Rationale for assuming equal group priors:** Does the assumption $q(s) = p(s), \forall s$ amount to saying e.g. that the proportion of male and female samples would be the same across all sub-regions (e.g. across all age groups)? If so, is that a reasonable assumption to make? I think the authors need to better explain how their choice of uncertainty set aligns with practical applications where one may want to enforce the form of local fairness they describe.

(4) **Distribution shift in Sec 4.2:** In the experiments with the Adult/census distribution shift in Sec 4.2, do we have reasons to believe that uncertainty set used would capture the "real-world" distribution shift considered? I am again particularly curious about the relevance of the assumption $q(s) = p(s)$ here.

---

> ### Author Response · Authors · 2023-11-17
>
> > (1) Complexity of problem for formulation: The authors seek to impose a robust version of the demographic parity constraint in (3), by introducing a maximization over an uncertainty set in the constraint. Instead of directly seeking to solve this problem, they formulate an equivalent problem with an additional argmin on an adversary's reconstruction loss (4). Could authors elaborate why they chose to work with (4) instead of the simpler form in (3)? It appears that the trick of applying instance-level weighting function could be applied as well to solving (3).
>
> Thank you for your question. We chose to work with (4) because it allows us to introduce an adversary’s reconstruction loss and make the problem differentiable, which greatly helps optimization, specifically when working with neural classifiers. Directly solving (3) would be intractable as the function $\hat{f}$ is non-differentiable after thresholding ($\hat{f}_{w_f}(x)=\mathbb{1}_{f_{w_f}(x)>0.5}$), making it unsuitable for gradient-based optimization. Our approach is inspired by the fairness reconstruction in adversarial settings, as demonstrated by Zhang et al. (2018), which also propose to enforce fairness by mitigating the ability of an adversary to reconstruct the sensitive attribute S from the probability f(X). Theoretical justification of this kind of adversarial approach for fairness is given in the second paragraph of Appendix A.2.2.
>
>
>
> > (2) Assumptions on $q$: Could the authors formally state the assumptions on the class of perturbed distributions $q$, preferably early on in the paper (e.g. Sec 2).
>
> In section 2, we give an overview of the problem, by giving a general formulation (eq 3) and then reviewing some related works that attempt to address fairness with DRO approaches. In section 2.3, we review different choices for the considered uncertainty set ${\cal Q}$, which has a strong impact on the behavior of the methods in practice. We indeed see from eq (3) that if ${\cal Q}$ is too large or far from $p_{test}$ (the target distribution for the classifier), the process might lead too overconstrain  the prediction, with the consideration of fairness for irealistic groups, which results in poor accuracy for the target classifier. Thus, for distribution shifts purposes followed by related works, the main requirement for ${\cal Q}$ is that it contains $p_{test}$, and is tight around it. Assuming that $p_{test}$ is close to $p$, this is done for instance by a KL constraint w.r.t. $p$.
>
> For our local fairness objective, we also want that the discrepancies of $q$ w.r.t. $p$ are smooth in the feature space, so that the fairness constraint does not increase mitigation on specific disconnected individuals, but rather on local areas of the space, which would be somehow ignored in classical global fairness approaches.
>
> Following the reviewer's recommendation, we added this last sentence in section 2.3 to strengthen the definition of our requirements regarding ${\cal Q}$. However, from our point of view, discussions about our validity constraints and the implied assumptions can difficultly be discussed here, for the reasons detailed below.
>
> > My understanding is that the group priors are required to be the same for both $q$ and $p$, i.e. $q(s) = p(s)$.
>
> The constraint of the initial problem from Eq.(3) compares expectations over prediction outputs for both sensitive groups, without any dependency on a given $q(s)$. $q$ only acts conditonnally on the binary sensitive value $s$ (the max is on posteriors $q(x|s)$).
>
> In its transposition in an adversarial formulation that is more tractable (eq.5), each sample is considered via an individual loss. It could be tempting to act on $q(s)$ to induce more powerful perturbations, but we show theoretically in Appendix A2.2 that it would induce a shift of the optimum, supported by empirical results in Appendix A2.3, by allocating higher $r$ weights to the most populated group (we added a new figure 6 in section A.2.3), to emphasize that most of the fairness effort is supported by the most populated group in this setting).  In order to avoid over-mitigate one subpopulation of the sensitive compared to the other one, we thus proposed to add an additional constraint to ensure that the marginal $q(s)$ equals the observed prior $p(s)$ through the definition of $r$.

---

> > ### Author Response · Authors · 2023-11-17
> >
> > > However, its unclear if the validity constraint described in Sec 3.1 would include all distributions $q$  for with the same group priors as $p$.
> >
> > In appendix 2.1, we show that the validity constraints that we consider actually ensure that $r$ implicitly defines a valid $q$ distribution, with equal priors $p(s)=q(s)$.
> >
> > For completeness, we could indeed show the reciprocal, that states that setting $r$ such that $p(s)=q(s)$ implies the considered constraints on $r$,  which would indicate that these "validity constraints include all distributions $q$ for with the same group priors as $p$".
> >
> > To do so, let us consider that $\forall s, p(s)=q(s)$.
> > Thus, we have $p(s)=\int q(x,s) dx=\int p(x|s) \frac{q(x,s)}{p(x|s)} dx$.
> > It directly follows that $\forall s, \int p(x|s) \frac{q(x,s)}{p(x|s)p(s)} dx = \mathbb{E}_{p(x|s)}[r(x,s)]=1$.
> >
> > Considering our validity constraints on $r$ is thus equivalent to set $q(s)=p(s)$ for every $s$ if dealing with an explicit distribution $q$. We added this in the end of appendix 2.1.
> >
> >
> > > (3) Rationale for assuming equal group priors: Does the assumption $q(s) = p(s), \forall s$ amount to saying  e.g. that the proportion of male and female samples would be the same across all sub-regions (e.g. across all age groups)? If so, is that a reasonable assumption to make? I think the authors need to better explain how their choice of uncertainty set aligns with practical applications where one may want to enforce the form of local fairness they describe.
> >
> > We respectfully want to point that there may be a misunderstanding on that point. This constraint does not impose anything on the subgroups, as $q$ is technically not directly the subgroups but rather the (weighted) global population ($q=rp$). The behavior suggested by the reviewer, i.e. “imposing equal proportion of male-female in all subregions”, which is, as pointed out, totally undesirable, might have been achieved by imposing a constraint such as: $p(s|x) = q(s|x)$. Again, this is not desirable in the context of this paper, and therefore was not considered.
> >
> > > (4) Distribution shift in Sec 4.2: In the experiments with the Adult/census distribution shift in Sec 4.2, do we have reasons to believe that uncertainty set used would capture the "real-world" distribution shift considered? I am again particularly curious about the relevance of the assumption $p(s)=q(s)$ here.
> >
> > Following our previous answers, we are interested in a generalization to a local proportion shift rather than a global sensitive shift. As explained previously, although we impose $q(s)=p(s)$ variations on $q(s|x)$ are still allowed. Therefore, the question of whether the uncertainty set captures these distribution shifts depends on the KL constraint rather than on the validity one.
> > We hope these answers help clarify the need and benefits of our validity constraint.

---

> ### Comment · Reviewer_bLrb · 2023-11-22
> **Thanks for the detailed response: follow-up questions**
>
> I thank the authors for the detailed response and clarifications, and for the revisions to the paper.
>
> I have a few follow-up questions:
> -  **Rationale for assuming equal priors**: Thanks for the clarification.  I am still not entirely convinced about the choice of uncertainty set. If "s" is a sensitive attribute that defines gender (for simplicity, male and female), am I correct that you consider perturbations to the data distribution that maintain the same **overall** proportion of male and females? If so, why are we only considering perturbations of this form? Wouldn't it natural to also expect data drifts that change the overall proportion of male and females?
>
> - **Differentiability of (3):** I understand that formulation (4) allows for differentiability of the objective. Couldn't the same be achieved by introducing a smooth surrogate objective for the non-differentiable indicators? Much of the initial works on group fairness employ some form of surrogate approximations to handle non-differentiable constraints (e.g. Zafar et al. (2015), Cotter et al. (2019)).
>
> P.S. Regarding my earlier point about the condition q(s) = p(s) imposing a constraint of equal priors across all sub-regions, let me be a bit more clear. When you compute the worst-case loss over all distributions q for which q(s) = p(s), you also include within this set distributions q that have *uniform density over samples in a particular age group*, and zero density on other samples. Such distributions define a sub-region of your data, but also additionally satisfy the condition q(s) = p(s) within the sub-region.

---

> > ### Author Response · Authors · 2023-11-23
> >
> > We thank the reviewer for the new follow-up questions. We hope these answers will help clarify the need and benefits of our validity constraint.
> >
> > > Rationale for assuming equal priors
> >
> > First of all, we would like to restate that local fairness is the primary goal pursued by our approach, although we show that it also allows to deal effectively with distributional drifts as an added benefit. For ensuring fairness, we show theoretically in A.2.2 and empirically in A.2.3 that letting $q(s)$ diverge from $p(s)$ induces an unbalance of the fairness effort, and thus does not align with our main objective of achieving local fairness. This is particularly visible in Figure 6 in the appendix using the Law datasets, where performance are strongly deteriorated without this constraint.
> >
> > While our conditional validity constraint that implies $q(s)=p(s)$ may look as limitative in case of distributional drifts (notably implying a drift of $p(s)$), we claim this is not the case as it does not constrain distributions inside both sensitive groups.  First, we want to point out that distribution drift on $p(s)$ without change on the support (i.e., with a drifted distribution $p_{test}$ that respects the absolute continuity hypothesis of $q$ w.r.t. to $p_{test}$) does not have any impact on the fairness of a **fully locally fair model**. It may only impact accuracy, which could be dealt with a second reweighting scheme for the prediction loss, same manner as done in classical DRO works such as [Michel et al. 2022] (this was not considered in our work to avoid over-complexity of presentation and highlight our innovation, but we acknowledge the potential interest and value of this complementary direction).
> >
> > In case of models where subgroups do not ensure $P(\hat{Y}=1|S)=P(\hat{Y}=1)$ (i.e. when the model is **not fully locally fair**), distributional drift may exhibit these local mistreatments in the test distribution (drift may make some "locally unfair" subgroups more represented in the test set), which explains the effectiveness of our approach in such settings (i.e., local fairness implies better robustness regarding distributional drifts). On the other hand, drifting $p(s)$, which corresponds to a global drift, is not supposed to amplify disparate mistreatments if the fairness effort is well balanced over both sensitive groups (as shown in our new figure 6 when using our conditional validity constraint). This is verified in our experiments corresponding to figure 3, for which a drift on $p(s)$ is observed (from $66\%$ male in train to $53\%$ in 2014 and 2015 test sets), without limiting the effectiveness of our approach.
> >
> >
> > We added this discussion in appendix A.2.3. Thanks again to the reviewer for this question that helped us improve the strength of our paper.
> >
> >
> > > Differentiability of (3): I understand that formulation (4) allows for differentiability of the objective. Couldn't the same be achieved by introducing a smooth surrogate objective for the non-differentiable indicators? Much of the initial works on group fairness employ some form of surrogate approximations to handle non-differentiable constraints (e.g. Zafar et al. (2015), Cotter et al. (2019)).
> >
> >
> > We acknowledge that using other formulations, such as the ones suggested by the reviewer, represents an interesting idea for future works. We have chosen the adversarial approach, inspired from [Zhang et al.], since it has generally been shown to outperform empirically other methods, including some of the ones mentioned by the reviewers (see for instance [1, 2] for extensive discussions and experiments on this topic). However, we also believe that  some other directions of extension of our work such as using a diferentiable Mutual Information metric [3] could be interesting to explore with a robust fair ratio. We have added a new comment on this topic in the conclusion.
> >
> >
> >
> > [1]  Adversarial mitigation to reduce unwanted biases in machine learning [Grari 2022]
> >
> > [2] One-Network Adversarial Fairness [Adel et al. 2019]
> >
> > [3] Learning Unbiased Representations via Mutual Information Backpropagation [Ragonesi et al. 2020]

---

> > > ### Author Response · Authors · 2023-11-23
> > >
> > > > Uniform density over samples in a particular age group, and zero density on other samples
> > >
> > >
> > > Regarding the last remark related to a set $\cal Q$ that would include distributions that allocate all probability mass solely to a specific sub-group of the dataset while respecting $p(s)=q(s)$,
> > > we would like to point out that such extreme cases, are prevented by our additional shape constraint.
> > > This  constraint, using a KL divergence, imposes that the distributions $q$ lie not far from $p$. This tends to prevent distributions such as the one described by the reviewer. As explained in section 3.1 in red, the hyperparameter $\tau$ is  strongly related to implicitly tuning the size of the smallest subgroup of the population for which we ensure fairness. If $\tau=0$, then ROAD approach might focus on too small groups (e.g. even a unique individual for extreme cases), which can lead to constant classifiers, as reported in [Martinez et al., 2021] (c.f., section 2.3). We therefore seek at defining a good tradeoff by setting a $\tau$ value that well balances between enough freedom to focus on specific unfair sub-regions, and sufficient regularization that avoids pathologic sharp distributions on too small sets of individuals.

---

### Official Review · Reviewer_d8Gj · 2023-11-04

**Soundness:** 3 good
**Presentation:** 3 good
**Contribution:** 3 good
**Rating:** 6
**Confidence:** 4

**Summary:**

This paper addresses a practical issue in methods that ensure group fairness in machine learning models. The methods that aim for group fairness can often be unfair to a subpopulation. To this end, authors combine adversarial learning and distributionally robust optimization (DRO) to learn globally fair classification models. DRO reweighs the sample, assigning more weights to samples that are easier to adversary. This ensures that fairness constraints would hold on even worst-case subgroups. Notably, the proposed method does not use subgroup labels. They propose two methods for reweighing the samples --- One exploits the loss of adversary to compute the weights (BROAD), and the other uses a separate neural network (which is optimized alongside during training) to compute the weights (called ROAD).

The authors performed experiments on three fairness datasets. They consider worst-case demographic parity on subgroups as a fairness constraint in one scenario. In the other scenario, they learn the fair model under distribution shift and use equalized odds as the fairness measure. In both cases, they show that ROAD provides a better trade-off between accuracy and fairness than other methods.

**Strengths:**

- The paper addresses an important practical issue in group fair learning approaches and is a step towards ensuring the reliability of fair learning methods.
- ROAD is consistently better than the other approaches for different levels of fairness and all datasets. In particular, I like using trade-off curves to report the results.
- The setting does not require specifying subgroups during training and can be fair w.r.t. to any choice of subgroup due to worst-case distribution consideration. To this end, combining DRO with adversarial learning is interesting.

**Weaknesses:**

- **Baseline**: In fig 3 and 4, some of the baselines do not span all ranges of fairness. Why do we see this behavior? Are the baselines tuned correctly?
- The proposed approach can be tricky to implement with several additional hyperparameters and the adversarial nature (min-max optimization) of the problem.
- **Results**: The experiments and reported results could be more elaborate:
  - How does DRO affect global fairness compared to other baselines? Fig 3 only reports local fairness in the form of the worst DI. However, the global fairness results are not reported.
  - While extending this framework to EO measure is straightforward, results with EO and local fairness are not reported. Similarly, DI and distribution shift results would strengthen confidence in the method.

**Questions:**

- **ROAD vs BROAD**: It is quite unintuitive that BROAD, an exact solution, performs poorly than ROAD, which uses a neural network to predict weights. The authors argued that ROAD is better because of the smoothing effects. Could it be that what is easy and hard for adversaries may change rapidly (or adversaries' output may be noisy) due to noisy batch gradients, and using an NN in ROAD may just be smoothing that out? In that case, would it make sense to use an EMA-like estimate with BROAD?

- Fig 4 (Left Fig): Are these unnormalized values? Isn't weight, i.e., $r$, positive and less than 1?
- **Adapting the method for EO**:
  - **Parameterization of weight network:** If we use BROAD for EO, it will use the adversary's loss, which uses labels $y_i$. Thus $r_i$ is a function of $x_i, s_i$ and $y_i$. Should the weight network (in ROAD) also use $y_i$ to predict r$?
  - **Validity constraint:** Would the validity constraint change for EO? That is, would it require normalization over both $s_i$ and $y_i$ or only $s_i$ as explained in the paper?

---

> ### Author Response · Authors · 2023-11-17
>
> We thank the reviewer for the valuable comments. We address the various questions/concerns below.
>
> >  Baseline: In fig 3 and 4, some of the baselines do not span all ranges of fairness. Why do we see this behavior? Are the baselines tuned correctly?
>
> We assume that by “fig 3 and 4”, the reviewer instead meant fig 2 and 3.
> In these figures, some curves indeed do not span all ranges of fairness. This is due to the fact that some methods do not offer sufficient control, depsite an extensive sweeping of their hyper-parameter domains via greed search,  that would enable to reach points in some areas of the support. For fig 2, this is accentuated by a filter that removes points that do not respect a minimal threshold of global fairness (as we are interested in inspecting local fairness for globally fair models). For fig 3, this is emphasized by distribution drifts: while for in-distribution test set (in the leftmost curves), every method allows a wide span of the global fairness support, results for other plots are impacted by drifts that make pareto fronts reduced on few points dominating every others for some methods.
>
>
> > How does DRO affect global fairness compared to other baselines? Fig 3 only reports local fairness in the form of the worst DI. However, the global fairness results are not reported.
>
> In fact, that figure reports results of local fairness for a high requirement for global fairness (global DI < 0.05). We see that our approach achieves this global fairness level (while some approaches such as fairLR on COMPAS have no point in the plot since never satisfying this condition), with competitive levels of accuracy for any local fairness score.
>
> > While extending this framework to EO measure is straightforward, results with EO and local fairness are not reported. Similarly, DI and distribution shift results would strengthen confidence in the method.
>
>
> You are right. We launched experiments to complement this. We have added a new section in appendix "A.6 Additional results on the Local Equalized-Odds criterion" (page 20). You may find these results in Figure 12 in the current version of the paper. The local unfairness is here measured using worst Disparate Mistreatment Rate (worst-1-DMR), that we defined by adding worst-1-FPR and worst-1-FNR.
>
>
> >  ROAD vs BROAD: It is quite unintuitive that BROAD, an exact solution, performs poorly than ROAD, which uses a neural network to predict weights. The authors argued that ROAD is better because of the smoothing effects. Could it be that what is easy and hard for adversaries may change rapidly (or adversaries' output may be noisy) due to noisy batch gradients, and using an NN in ROAD may just be smoothing that out? In that case, would it make sense to use an EMA-like estimate with BROAD?
>
> Thank you for this insightful question regarding the difference in performance between ROAD and BROAD.
>
> The problems that may hinder BROAD performance go beyond smoothing. In the DRO literature, it is known, and deeply discussed (see [1, 2, 3, 4] for instance), that the non-parametric solution (such as BROAD) may lead to suboptimal performance compared to parametric ones. This may come from several causes, such as the fact it is essentially solving a problem that is fundamentally too pessimistic: i.e., the uncertainty set $\mathcal{Q}$ may include distributions that are too difficult to learn, and not necessarily representative of real-world constraints. Thanks for the suggestion of using an EMA-like estimate with BROAD, which could alleviate learning unstability issues, but might still be limited by the use of individual weights: in BROAD, weights of training samples are only interlinked via the outputs from the classifier, hence at the risk of conflicting with our notion of local fairness (which is defined on the feature space).
>
> On the other hand, the parametric, neural network-based, approach (such as ROAD), “has much less (only parametric) freedom to shift the test distribution compared to the adversary that uses non-parametric weight” (sic) [1]. The lipschitzness of neural networks can add additional implicit locality smoothness assumptions, thus helping define distributions $q$ as subregions of the feature space. In classical DRO, this leads to models that are robust to more meaningful distribution shift. In our setting of DRO for local fairness, this also helps focusing on local fairness with groups formed with similar individuals. As discussed in our paper, the network architecture defines the level of local smoothness of considered groups. And a network of infinite capacity that completes training would have, in theory, the same behavior as BROAD.
>
> [1] Hu, Weihua, et al. “Does distributionally robust supervised learning give robust classifiers?.” International Conference on Machine Learning. PMLR, 2018.
> [2] Duchi et al. 2020 Distributionally Robust Losses Against Mixture Covariate Shifts. ROAD is a heuristic approximation to the objective

---

> ### Author Response · Authors · 2023-11-17
>
> [2] Duchi et al. 2020 Distributionally Robust Losses Against Mixture Covariate Shifts. ROAD is a heuristic approximation to the objective
>
> [3] Michel et al. 2021 Modeling the Second Player in Distributionally Robust Optimization
>
> [4] Michel et al. 2022 Distributionally Robust Models with Parametric Likelihood Ratios
>
>
> >  Fig 4 (Left Fig): Are these unnormalized values? Isn't weight, i.e., $r$  positive and less than 1?
>
> Normalization is on the mean, not the sum: we impose $\mathbb{E}_p[r(x,s)]=1$ to ensure that $q$ is a distribution that respects $\int q(x,s)=1$. As we implement this constraint with an unbiased estimator based on training samples (i.e., $\frac{1}{n} \sum r(x_i,s_i)=1$), we get that $r$ values sum to $n$, not $1$. This can be seen for instance in our implementation of BROAD where the partition function is divided by $n$.
>
> >  Parameterization of weight network: If we use BROAD for EO, it will use the adversary's loss, which uses labels $y_i$. Thus $r_i$ is a function of $x_i$, $s_i$ and $w_i$. Should the weight network (in ROAD) also use $y_i$  to predict $r_i$?
>
> Yes, we agree with the reviewer and confirm that in order to optimize for EO, $r$ needs to be a function of $x$, $s$ and $y$; and that the weight network uses $y$ to predict $r$. We have provided more details about this in the appendix of the paper (please see A.10: "ADAPTING ROAD TO EQUALIZED ODDS").
>
> >  EO, validity constraint: Would the validity constraint change for EO? That is, wuld it require normalization over both $s_i$ and $y_i$, or only $s_i$ as explained in the paper?
>
> Thank you for your question. Following similar reasoning as for our validity constraint in the case of DP (section A2.2), we have indeed also considered adapting the validity constraint for EO by conditioning it on $y$. This effectively means imposing the following constraints:
> $\forall s_i, \forall y_i, \mathbb{E}_{x | s=s_i, y=y_i}r(x,s,y)=1$
> However, depending on the dataset, conditioning the batches on both $s$ and $y$ may lead to very small subsamples,  i.e. non-significant expectation estimates. As a result, the obtained models were sometimes less robust, and this lead us to not pursuing this direction any further. We have added a discussion on this in Section A.7 with an experiment comparing the two constraints (Figure 13).

---

### Author Response · Authors · 2023-11-17
**Global comment**

We thank the reviewers for their valuable comments and questions. Besides our answers, which can be found below, these insights have lead us to bring several modifications to the paper, which we believe have lead to substantially improving its quality. The modifications have been directly integrated to the updated version of the pdf, in red. Here is a list:

- Figure 12, and more generally Section A.6: Local Fairness experiment for EO (suggested by Reviewer 1)
- Section A.10 Adapting ROAD to Equalized Odds + Figure 13: Experimenting with the validity constraint when optimizing for EO (following a question by Reviewer 1)
-  Section 2.3: Clarifications and motivations on the uncertainty set (following most questions and comments by Reviewer 2)
-  Appendix A.2.1: Clarifications on the implications of the equality of the priors (following questions by Reviewer 2)
- Figure 6 in Section A.2.3: Empirical results showcasing the need for the equality on the priors (following questions by Reviewer 2)
- References on applied fairness, related works on adversarial learning, and bias in ML (suggestions by Reviewer 3)
- Appendix A.11: Discussion on limitations of the notion of local fairness and the proposed approach (suggestion by Reviewer 3)
- Sections 2.2, 2.3 and 3: Discussions on regularization and priors regarding subregion splits. (question by Reviewer 4)
- Figure 11, and more generally Appendix A.5: Empirical results and and clarifications on the difference between our proposition and [Michel et al. 2022] (question by Reviewer 4)

---

### Meta-Review · Area_Chair_HrWr · 2023-12-12

**Metareview:**

This work combines a number of promising directions in the fairness/robustness space to debiasing, specifically assuming a very strong adversary as proxy for the same minmax/maxmin operation that is encoded in many group fairness constraints/regularizers.  Reviewers appreciated the motivation (w.r.t. pushing an active line of research in the ML/fair ML community forward), approach, and experimental results.

**Justification For Why Not Higher Score:**

Rebuttals were acknowledged by reviewers and, in some cases, led to an active back-and-forth with the authors.  Reviewers largely kept their review scores the same, both from an upside and downside risk point of view, so -- coupled with some of the potential flaws of the work -- I see this a low variance / low risk acceptance of a strong but not outstanding paper.

**Justification For Why Not Lower Score:**

See above; scores, back-and-forth, validation of paper, etc.  It's a solid paper within ICLR's scope, shouldn't be rejected.

---

### Decision · Program_Chairs · 2024-01-16

Accept (poster)